# M3C: A Framework towards Convergent, Flexible, and Unsupervised Learning of Mixture Graph Matching and Clustering

**Jiaxin Lu**[1†], **Zetian Jiang**[2†], **Tianzhe Wang**[2], **Junchi Yan**[2*]

[1]Department of Computer Science, University of Texas at Austin

[2]Department of Computer Science and Engineering, Shanghai Jiao Tong University

`lujiaxin@utexas.edu`, `{maple_jzt,usedtobe,yanjunchi}@sjtu.edu.cn`

## Abstract

Existing graph matching methods typically assume that there are similar structures between graphs and they are matchable. This work addresses a more realistic scenario where graphs exhibit diverse modes, requiring graph grouping before or along with matching, a task termed mixture graph matching and clustering. Specifically, we introduce Minorize-Maximization Matching and Clustering (M3C), a learning-free algorithm that guarantees theoretical convergence through the Minorize-Maximization framework and offers enhanced flexibility via relaxed clustering. Building on M3C, we further develop UM3C, an unsupervised model that incorporates novel edge-wise affinity learning and pseudo label selection. Extensive experimental results on public benchmarks demonstrate that our method outperforms state-of-the-art graph matching and mixture graph matching and clustering approaches in both accuracy and efficiency.

## 1 Introduction

Graph matching (GM) (Yan et al., 2020b) constitutes a pervasive problem in computer vision and pattern recognition, with applications in image registration (Shen & Davatzikos, 2002), recognition (Duan et al., 2012; Demirci et al., 2006), stereo (Goesele et al., 2007), 3D shape matching (Berg et al., 2005; Petterson et al., 2009), and structure from motion (Simon et al., 2007). GM involves finding node correspondences between graphs by maximizing affinity scores, commonly formulated as the quadratic assignment problem (QAP), often referred to as Lawler's QAP (Loiola et al., 2007):

$$\mathbf{X} = \arg\max_{\mathbf{X}} \text{vec}(\mathbf{X})^\top \mathbf{K} \text{vec}(\mathbf{X}) \quad s.t. \ \mathbf{X} \in \{0,1\}^{n_1 \times n_2}, \mathbf{X}\mathbf{1}_{n_2} \leq \mathbf{1}_{n_1}, \mathbf{X}^\top \mathbf{1}_{n_1} \leq \mathbf{1}_{n_2} \quad (1)$$

Here, $\mathbf{X}$ is a permutation matrix encoding node-to-node correspondences, and $\mathbf{1}_n$ is an all-one vector. The inequality constraints accommodate scenarios with outliers, addressing the general and potentially ambiguous nature of the problem. Multiple graph matching (MGM) (Yan et al., 2015a;b; Jiang et al., 2021) extends GM by enforcing cycle consistency among pairwise matching results. GM and MGM are both NP-hard, leading to the proposal of approximate algorithms, either learning-free (Yan et al., 2016) or learning-based (Yan et al., 2020a).

In GM, whether in two-graph matching or multi-graph matching, a common assumption prevails: graphs must belong to the same category, and labels for both graphs and nodes are required. However, labeling can be costly, especially in domains like molecular design or drug discovery requiring domain-specific knowledge. Real-world scenarios involve mixtures of different graph types, e.g., in traffic tracing, frames may contain people, bicycles, and cars simultaneously. Matching with mixed graph types is a practical challenge in its nascent stage. In this paper, we introduce Mixture Graph Matching and Clustering (MGMC), aiming to align graph-structured data and simultaneously partition them into clusters, as shown in Fig. 1. This task seeks to mutually optimize both matching and clustering problems, thereby enhancing the outcomes of both tasks: matching establishes a similarity metric for clustering, while cluster information improves the results of intra-cluster matching.

Recent studies have explored mixture graph matching and clustering in two works: Decayed Pairwise Matching Composition (DPMC) (Wang et al., 2020b) and the graduated assignment neural

---

*Correspondence author. † denotes equal contribution. The SJTU authors were in part supported by NSFC (92370201, 62222607) and STCSM (22511105100).

Table 1: Comparison of the existing three works designated for mixture graph matching and clustering: DPMC (Wang et al., 2020b), GANN (Wang et al., 2020a), and ours: M3C/UM3C.

| Methods | DPMC | GANN | M3C/UM3C (ours) |
|---|---|---|---|
| Optimization Space | Discrete | Continuous | Discrete |
| Framework | - | Graduated Assign | Minorize-Maximization |
| Supergraph Structure | discrete tree | real-value edge fully connected | discrete and partly-connected |
| Joint Optimize | × | ✓ | ✓ |
| Convergence | × | slow | fast |
| Affinity Learning | - | node | node and edge |
| Pseudo Label | - | unselected | selected w/ relaxed indicator |
| Empirical Robustness | mediocre | hardly work given 2+ outliers | 30% accuracy improvement over GANN |

network (GANN) (Wang et al., 2020a). However, they suffer from certain drawbacks that warrant attention: 1) **Convergence**. DPMC exhibits convergence instability, while GANN has slow convergence. 2) **Rigid Structure**. DPMC relies heavily on its tree structure, and GANN tends to converge to a sub-optimal due to its hard clustering nature. 3) **Robustness**. GANN's robustness is compromised by noise, as shown in our experiments, where matching accuracy drops significantly with just two outliers and further deteriorates with more outliers.

We propose our solution, **M**inorize **M**aximization **M**atching and **C**lustering (**M3C**). M3C enjoys convergence guarantees and is based on a convergent alternating optimization solver. We utilize the cluster indicator from hard clustering to represent the discrete structure used for optimization, providing better information for graphs of different modes while preserving the convergence guarantee of the algorithm. We additionally introduce **UM3C**, which integrates the learning-free solver into an unsupervised pipeline, incorporating edge-wise affinity learning, affinity loss, and a pseudo-label selection scheme for higher robustness. A comprehensive comparison with previous works is summarized in the appendix Sec. B with Table 1. The main contributions of this paper are:

- We present M3C, a learning-free solver for the mixture graph matching and clustering problem, that guarantees convergence within the Minorize-Maximization framework, enhanced by a flexible optimization scheme enabled by the relaxed cluster indicator. This marks the first theoretically convergent algorithm for MGMC, to the best of our knowledge.

- We enhance M3C by integrating it with an unsupervised pipeline UM3C. Edge-wise affinity learning, affinity loss, and pseudo label selection are introduced for learning quality and robustness.

- M3C and UM3C outperform state-of-the-art learning-free and learning-based methods on mixture graph matching and clustering experiments. UM3C even outperforms supervised models BBGM and NGM, establishing itself as the top-performer for MGMC on public benchmarks.

## 2 RELATED WORKS

**Graph Matching** Graph matching has gained attention recently, with various techniques explored, including spectrum, semi-definite programming (SDP), and dual decomposition (Gold & Rangarajan, 1996; van Wyk & van Wyk, 2004; Cho et al., 2010; Tian et al., 2012; Egozi et al., 2012; Swoboda et al., 2017; Xu et al., 2019; Zhang et al., 2019; Liu et al., 2021). Multiple graph matching (MGM) introduces cycle consistency as regularization to encourage matching transitivity, whose methods fall into two categories: matrix factorization-based (Kim et al., 2012; Pachauri et al., 2013; Huang & Guibas, 2013; Zhou et al., 2015; Chen et al., 2014; Leonardos et al., 2017; Hu et al., 2018; Swoboda et al., 2019) and supergraph-based approaches (Yan et al., 2015b;a; Jiang et al., 2021; Wang et al., 2020b). The former enforces cycle consistency through matrix factorization, connecting all graphs with a universe-like graph for global consistency. The latter iteratively updates two-graph matchings by considering metrics along the supergraph path. Recent studies explore deep learning methods for feature extraction and learning-free or neural network solvers for matching (Zanfir & Sminchisescu, 2018; Wang et al., 2019; 2021; Wang et al., 2020; Rolínek et al., 2020; Yu et al., 2021; Wang et al., 2020a; Liu et al., 2020; Nurlanov et al., 2023; Fey et al., 2020; Jiang et al., 2022b; Lin et al., 2023; Jiang et al., 2022a), covering both supervised and unsupervised learning pipelines.

**Graph Clustering** In this paper, we tackle graph clustering, which aims to group similar graphs. One approach involves embedding each graph into a Hilbert space and using clustering methods like k-means (Xu & Lange, 2019) Previous work (Wang et al., 2020b;a) commonly use Spectral Cluster (Ng et al., 2002) based on pairwise affinity scores. Another approach (Hartmanis, 1982; Poljak & Rendl, 1995; Trevisan, 2012; Goemans & Williamson, 1995) utilizes max cut (De La Vega & Kenyon, 2001; Festa et al., 2002; Poljak & Rendl, 1995; Trevisan, 2012; Goemans & Williamson,

1995), treating input graphs as nodes in a supergraph and assigning weights to edges based on pairwise scores. Alternative formulations for graph clustering include min cut (Johnson et al., 1993), normalized cuts (Xu et al., 2009), and multi-cuts (Kappes et al., 2016; Swoboda & Andres, 2017).

**Mixture Graph Matching and Clustering**    Matching with mixtures of graphs entails finding node correspondence and partitioning graphs into clusters with works: DPMC (Wang et al., 2020b) and GANN (Wang et al., 2020a). GANN introduces GA-MGMC, a graduated assignment-based algorithm optimizing the MGMC problem in a continuous space, followed by projecting results to discrete matching. DPMC, a learning-free solver, constructs a maximum spanning tree on the supergraph and updates matching along the tree. Another work (Bai et al., 2019) embeds the input graph for matching into an embedding vector for graph clustering. Joint matching and node-level clustering are explored (Krahn et al., 2021), solving node correspondence and segmenting input graphs into sub-graphs. This paper focuses on the joint graph matching and clustering problem, a relatively new area in the literature. It represents an advancement for more open settings.

## 3    BACKGROUND AND PROBLEM FORMULATION

We introduce some preliminary concepts, definitions, and the proposed problem formulation in this section. The definition of notations are introduced in Appendix Sec. A.

**Definition 1 (Matching composition).** *Matching composition involves combining pairwise matching results to enhance the initial matching configuration:* $\mathbf{X}^{t+1} = \mathbf{X}_{ik_1}^t \mathbf{X}_{k_1 k_2}^t \dots \mathbf{X}_{k_s j}^t$. *We further define the matching composition space of $\mathcal{G}_i$ and $\mathcal{G}_j$ to encompass all possible compositions between them* $\mathbf{P}_{\mathbb{X}}(i,j) = \{\mathbf{X}_{ik_1} \dots \mathbf{X}_{k_s j} | s \in \mathbb{N}^+, 1 \le k_1 \dots k_s \le N\}$.

**Definition 2 (Supergraph).** *Supergraph is a common protocol for describing multi-graph matching. The supergraph $\mathcal{S} = \{\mathcal{V}, \mathcal{E}, \mathbf{A}\}$ consists of vertices corresponding to graphs $\mathcal{V} = \{\mathcal{G}_1, \dots, \mathcal{G}_N\}$ and edges weighted by pairwise matching affinity scores $\mathbf{X}_{ij}$, with adjacency $\mathbf{A} \in \{0,1\}^{N \times N}$.*

*Each path on the supergraph corresponds to a matching composition. The weight of the path from $\mathcal{G}_i$ to $\mathcal{G}_j$ is defined as the affinity score of the matching composition: $\mathbf{X}_{ij} = \mathbf{X}_{ik_1} \mathbf{X}_{k_1 k_2} \dots \mathbf{X}_{k_s j}$. The matching composition space of $\mathcal{G}_i$ and $\mathcal{G}_j$ can be represented as all the paths from $\mathcal{G}_i$ to $\mathcal{G}_j$ on the supergraph:* $\mathbf{P}_{\mathbf{A}}(i,j) = \{\mathbf{X}_{ik_1} \dots \mathbf{X}_{k_s j} | \forall a_{ik_1} = \dots = a_{k_s j} = 1\}$.

**Definition 3 (Cluster indicator).** *The cluster indicator is defined to describe whether two graphs belong to the same cluster. It is represented by the cluster indicator matrix $\mathbf{C} \in \{0,1\}^{N \times N}$, where $c_{ij} = 1$ denotes the same class. The transitive relation $c_{ij} c_{jk} \le c_{ik}$ serves as the sufficient and necessary condition for $\mathbf{C}$ to be a strict cluster division (see proof in the appendix). The number of clusters can be determined by the number of **strongly connected components** of $\mathbf{C}$, namely $SCC(\mathbf{C})$.*

**Problem Formulation**    The MGMC problem can be formulated as a joint optimization problem, where matching results maximize pair similarity to facilitate clustering, while the cluster indicator guides matching optimization in turn. Given the set of pairwise affinity and number of clusters $N_c$, the overall objective $\mathcal{F}(\mathbb{X}, \mathbf{C})$ for joint matching and clustering can be written as follows:

$$\max_{\mathbb{X}, \mathbf{C}} \mathcal{F}(\mathbb{X}, \mathbf{C}) = \max_{\mathbb{X}, \mathbf{C}} \frac{\sum_{ij} c_{ij} \cdot \text{vec}(\mathbf{X}_{ij})^\top \mathbf{K}_{ij} \text{vec}(\mathbf{X}_{ij})}{\sum_{ij} c_{ij}} \tag{2}$$

$$s.t. \ \mathbf{X}_{ij} \mathbf{1}_{n_j} \le \mathbf{1}_{n_i}, \mathbf{X}_{ij}^\top \mathbf{1}_{n_i} \le \mathbf{1}_{n_j}, \quad c_{ik} c_{kj} \le c_{ij}, \forall i, j, k, \quad \text{SCC}(\mathbf{C}) = N_c.$$

where $\mathbb{X}$ represents the pairwise matching matrices, and $\mathbf{C}$ denotes the cluster indicator matrix. The first part of constraints ensures that $\mathbf{X}_{ij} \in \{0,1\}^{n_i \times n_j}$ is a (partial) permutation matrix, and the second part requires $\mathbf{C} \in \{0,1\}^{N \times N}$ to be a strict cluster division with $N_c$ clusters. The term $\frac{1}{\sum_{ij} c_{ij}}$ acts as a normalization factor to mitigate the influence of cluster scale. The cycle consistency within each cluster is either enforced as a constraint $\mathbf{X}_{ik} \mathbf{X}_{kj} \le \mathbf{X}_{ij}$, $\forall c_{ik} = c_{kj} = c_{ij} = 1$, or softly encouraged by the algorithm.

## 4    A LEARNING-FREE APPROACH: M3C

In this section, we present our learning-free algorithm, M3C. We start by converting the original problem into a Minorize-Maximization (MM) framework (Hunter & Lange, 2004; Mairal, 2015), a nontrivial achievement not realized before (Sec. 4.1). Additionally, we propose a relaxed indicator that allows for more flexible exploration by relaxing the hard constraints from independent clusters to the global and local rank of affinities (Sec. 4.2). We finally present the full algorithm (Sec. 4.3).

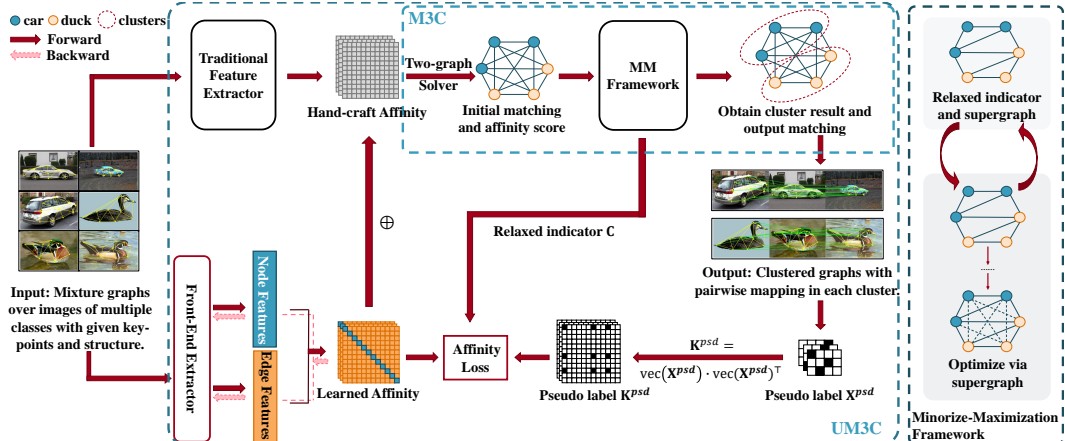

Figure 1: Pipeline of M3C and its unsupervised learning extension called UM3C. We use two clusters: ducks and cars as an illustration example. The node in the plot refers to the graph sample for matching and different colors refer to different clusters. M3C is a leaning free solver as shown in the dotted box in the top middle with a minorize-maximization framework built on a relaxed indicator (see Sec. 4). It is extended to an unsupervised pipeline (see Sec. 5) by learning the edge-wise affinity matrix $\mathbf{K}$ and a pseudo label selection scheme.

## 4.1 CONVERTING THE PROBLEM SOLVING INTO A MINORIZE-MAXIMIZATION FRAMEWORK

Recall that DPMC (Wang et al., 2020b) device an alternative system for matching and clustering, suffers non-convergence. To utilize the mutual optimization nature and guarantee convergence, we introduce a Minorize-Maximization (MM) framework.

We start by presenting a new objective with a single variable $\mathbb{X}$, denoted as $f(\mathbb{X}) = \mathcal{F}(\mathbb{X}, h(\mathbb{X}))$, to incorporate the MM framework into our approach. Here, $h(\mathbb{X}) = \arg\max_{\mathbf{C}} \mathcal{F}(\mathbb{X}, \mathbf{C})$, represents the optimal cluster division for $\mathbb{X}$.

The MM framework works by finding a surrogate function $g(\mathbb{X}|\mathbb{X}_0) = \mathcal{F}(\mathbb{X}, h(\mathbb{X}_0))$, which minorizes the original objective function $f(\mathbb{X})$. By optimizing this surrogate function, we can iteratively improve the objective or maintain its value. The iterative steps are as follows:

- Construct the surrogate function $g(\mathbb{X}|\mathbb{X}^{(t)}) = \mathcal{F}(\mathbb{X}, h(\mathbb{X}^{(t)}))$ by inferring the best cluster based on the current matching results $\mathbb{X}^{(t)}$.

- Maximize $g(\mathbb{X}|\mathbb{X}^{(t)})$ instead of $f(\mathbb{X})$, which can be solved using graph matching solvers.

The above iteration guarantees that $f(\mathbb{X})$ is monotonic incremental:

$$f(\mathbb{X}^{(t+1)}) \geq g(\mathbb{X}^{(t+1)}|\mathbb{X}^{(t)}) \geq g(\mathbb{X}^{(t)}|\mathbb{X}^{(t)}) = f(\mathbb{X}^{(t)}). \tag{3}$$

On the other hand, $f(\mathbb{X})$ exhibits a natural upper bound, i.e. $f(\mathbb{X}) < \sum_{ij} \sum_{abcd} \mathbb{K}_{ij}(a, b, c, d)$. According to the Monotone Convergence Theorem, if a monotone sequence of real numbers, such as $f(X^t)$, is bounded, then the sequence converges. Details of the proof can be found in Sec. C.2.

## 4.2 RELAXATION ON CLUSTER INDICATOR

The proposed framework benefits from a convergence guarantee. However, subsequent theoretical analyses indicate that hard clustering tends to converge rapidly to a sub-optimal solution and lacks of the capability to correct the clustering results. A similar challenge is also faced by GANN (Wang et al., 2020a), which overlooks the intrinsic differences between matched pairs and assigns them clustering weights of either 1 or a constant $\beta$. Owing to its inherent hard clustering characteristic, GANN's performance exhibits high sensitivity to both parameter fine-tuning and the presence of outliers. This underscores our motivation to relax the hard constraints.

**Proposition 4.1.** *If the size of each cluster is fixed, the hard cluster indicator converges to the local optimum in one step:*

$$\mathbf{C}^{(t)} = \mathbf{C}^{(t+1)}, if \{N_{g_1}^{(t)}, \ldots, N_{g_{N_c}}^{(t)}\} = \{N_{g_1}^{(t+1)}, \ldots, N_{g_{N_c}}^{(t+1)}\}. \tag{4}$$

Please refer to Sec. C.3 for the detailed proof. Two key observations are: First, the cluster indicator attains a local optimum for each cluster size group. Second, it converges within one optimization step when the target cluster size is known. Such convergence fixes the optimization space for finding optimal matching compositions, constraining the exploration of both matching and clustering results. Experimental results confirming the quick convergence is provided in Appendix. H.4.

To overcome the constrained exploration, we propose relaxing the hard constraints on the original cluster indicator. We present two relaxations on the number of graph pairs for the new indicator $\hat{\mathbf{C}}$:

$$\sum_{ij} \hat{c}_{ij} = r \cdot N^2 \text{ (global constraint);} \quad \sum_{j} \hat{c}_{ij} = r \cdot N, \forall i \text{ (local constraint)} \quad (5)$$

Here, $r \in [0, 1]$ is a hyper-parameter that adjusts the ratio of chosen pairs.

The global version limits the total number of selected graph pairs, while the local version restricts pair numbers for each graph. We refer to this new approach as the **relaxed indicator**, which provides three advantages: 1) The relaxed indicator assesses each graph pair individually. Considering the potential errors in clustering results, loosening clustering constraints—disregarding cluster numbers and transitive relations $c_{ik}c_{kj} \leq c_{ij}$—enables the incorporation of valuable information from other clusters. This, in turn, provides greater flexibility in the optimization space. 2) It enhances pseudo label selection in unsupervised learning by selecting graph pairs with higher affinity scores, as discussed in Sec. 5. 3) Compared to GANN (Wang et al., 2020a), which also proposes a relaxation for **C**, our approach maintains discrete constraints, resulting in better convergence than GA-MGMC.

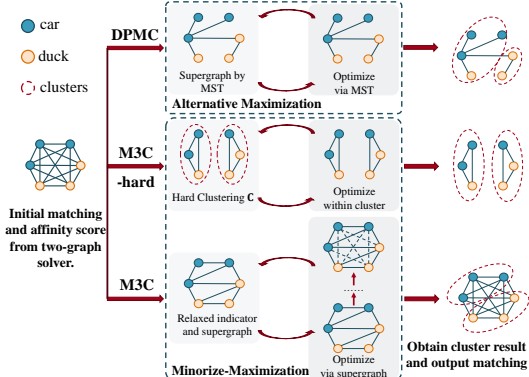

Figure 2: Three optimization structures. **DPMC**: rigid tree structure, no convergence guarantee. **M3C-Hard**: constrained exploration. **M3C**: relaxed indicator, better solutions with convergence.

### 4.3 THE CLUSTER INDICATOR RELAXED ALGORITHM: M3C

We introduce the M3C algorithm, which leverages the relaxed indicator $\hat{\mathbf{C}}$ within the MM framework, consisting of three key parts: initialization, surrogate function construction, and maximization step. The details are outlined in Alg. 1 in the appendix.

**Initialization** We obtain the initial matching $\mathbb{X}^{(0)}$ using a two-graph solver, such as RRWM (Cho et al., 2010), which is both cost-effective and aligns with existing literature on multiple graph matching (Yan et al., 2015b; Jiang et al., 2021).

**Surrogate Function Construction** With the introduction of relaxed constraints in Sec. 4.2, we first present two methods for solving the optimal relaxed indicator $\hat{\mathbf{C}}$ using the so-called global and local constraints, respectively.

$$\hat{\mathbf{C}}^{(t)} = \hat{h}(\mathbb{X}^{(t-1)}) = \arg\max_{\hat{\mathbf{C}}} \frac{\sum_{ij} c_{ij}}{rN^2} \text{vec}(\mathbf{X}_{ij}^{(t-1)})^\top \mathbf{K}_{ij} \text{vec}(\mathbf{X}_{ij}^{(t-1)})$$

$$s.t. \sum_{ij} c_{ij} = rN^2 \text{ (global constraint) or } \sum_{j} c_{ij} = rN, \forall i \text{ (local constraint).} \quad (6)$$

In Eq. 6, given $\mathbb{X}^{(t-1)}$ and $r$, the affinity scores and the normalization coefficients remain fixed. The objective is to select graph pairs with higher affinity scores. For the global constraint, we rank all graph pairs by affinity and set a threshold at the highest $rN^2$ values. For the local constraint, we rank the neighbors of each graph and select the top $rN$ pairs in the context of k-nearest-neighbor. These two algorithms are denoted as **global-rank** and **local-rank**, respectively.

Building upon the local and global schemes, we propose a fused approach named **fuse-rank**. It combines both global and local constraints by introducing a bi-level ranking. The **local** rank of a pair of graphs $(\mathcal{G}_u, \mathcal{G}_v)$ is determined as: $R_{uv} = i + j$, where $\mathcal{G}_u$ is the $i$-th nearest neighbor of $\mathcal{G}_v$,

and $\mathcal{G}_v$ is the $j$-th nearest neighbor of $\mathcal{G}_u$. Subsequently, we establish a **global** threshold across all graph pairs based on $\{R_{uv}\}$. This approach allows the induced relaxed indicator $\hat{\mathbf{C}}$ to reflect both local and global affinity relationships among graphs. We will evaluate the performance of the above three strategies in our experiments.

**Surrogate Function Maximization**   In hard clustering, the maximization step optimizes within clusters, which is trivial to apply MGM algorithms. However, with the relaxed indicator, the optimization structure changes and requires a modification of the algorithm. We first present our approach, and then resonate why it is a well-defined generalization to address this issue.

With the supergraph mentioned in Def. 2 and the relaxed indicator, we can construct an incomplete supergraph with edges connecting graphs of the same class. The adjacency $\mathbf{A}$ of this supergraph corresponds numerically to the relaxed indicator $\hat{\mathbf{C}}$, thus we simply set $\mathbf{A} = \hat{\mathbf{C}}$. Let $\mathbf{P}_{\mathbf{A}^{(t)}}$ denote all the paths (matching composition space) based on adjacency $\mathbf{A}$, different from traditional MGM problem, the optimization step becomes:

$$\mathbb{X}^{(t)} = \underset{\mathbf{X}_{ij} \in \mathbf{P}_{\mathbf{A}^{(t)}}(i,j)}{\arg\max} \sum_{ij} \mathrm{vec}(\mathbf{X}_{ij})^\top \mathbf{K}_{ij} \mathrm{vec}(\mathbf{X}_{ij}) \tag{7}$$

In Eq.7, the optimization space $\mathbf{P}_{\mathbf{A}^{(t)}}(i, j)$ increases with the number of paths between $\mathcal{G}_i$ and $\mathcal{G}_j$, making it more likely to optimize graphs of the same class. This property mirrors that of the cluster indicator and renders the maximization objective equivalent to Eq. 7 when given the relaxed indicator $\hat{\mathbf{C}}$. Furthermore, when the relaxed indicator degenerates to hard cluster division, the supergraph reduces to several connected components, causing the optimization to focus solely on graphs within the same cluster. Hence, Eq.7 is a well-defined generalization of the maximization objective.

With the proposed algorithm, the relaxed indicator $\hat{\mathbf{C}}$ is no longer fixed, expanding the optimization space of the maximization step. This enables the maximization step to enhance the quality of the relaxed cluster. Both steps work jointly, optimizing both clustering and matching results.

In summary, we have introduced a novel learning-free approach for joint matching and clustering with a convergence guarantee. In the following section, we aim to integrate deep neural networks into our framework to enable an unsupervised learning paradigm for graph matching and clustering.

## 5   Unsupervised Learning Model: UM3C

We introduce UM3C, an unsupervised extension to M3C, enabling joint matching and clustering within a guaranteed convergence framework. Recent deep learning advancements in graph matching (Zanfir & Sminchisescu, 2018; Wang et al., 2019; Rolínek et al., 2020) highlight the value of learning node and edge features. However, M3C's discrete optimization hinders gradient computation and machine learning integration. Previous attempts, like BBGM (Rolínek et al., 2020), struggled with gradient approximation and affinity generalization. GANN (Wang et al., 2021), although unsupervised, focused solely on Koopmans-Beckmann's QAP, ignoring edge features and outlier handling. UM3C addresses these challenges with two techniques: 1) edge-wise affinity learning and affinity loss, guided by pseudo matching label from M3C solver; 2) Pseudo-label selection using the introduced relaxed indicator to improve pseudo label quality. We detail these techniques in the following subsections and present M3C's integration into the unsupervised learning pipeline.

### 5.1   Edge-wise Affinity Learning

Our solver M3C adopts Lawler's QAP (Loiola et al., 2007), which embraces second-order information for enhanced performance. This requires meticulous design of the affinity matrix $\mathbf{K}$.

Recent research on graph matching (Rolínek et al., 2020; Wang et al., 2021) adopts deep learning pipelines to compute node features $\mathbf{F}_i^n$ and edge features $\mathbf{F}_{ij}^e$ using VGG-16 (Simonyan & Zisserman, 2014) and Spline CNN (Fey et al., 2018). A learned affinity $\mathbf{K}^{learn}$ can be constructed from these first and second-order features as: $\mathbf{K}_u^{learn} = (\mathbf{F}^n)^\top \mathbf{\Lambda} \mathbf{F}^n, \mathbf{K}_q^{learn} = (\mathbf{F}^e)^\top \mathbf{\Lambda} \mathbf{F}^e$. Here, $\mathbf{K}_u$ and $\mathbf{K}_q$ denote unary and quadratic affinities respectively, with $\mathbf{\Lambda}$ set to $\mathbf{I}$ for stable training.

Our goal is to improve $\mathbf{K}^{learn}$ using ground truth $\mathbf{X}^{gt}$ (or pseudo-label $\mathbf{X}^{psd}$). Commonly, a differentiable pipeline calculates prediction matching $\mathbf{X}$ for applying the loss function. However, this approach conflates affinity construction and the solver, leading to customized affinities limited to the training solver and hampered generalization.

We devise a cross-entropy loss to quantify the discrepancy between two input affinity matrices:

$$\mathcal{L}(\mathbf{K}^{learn}, \mathbf{K}^{gt}) = \sum_{pq} K_{pq}^{gt} \log(K_{pq}^{learn}) + (1 - K_{pq}^{gt}) \log(1 - K_{pq}^{learn}), \qquad (8)$$

where $\mathbf{K}^{gt} = \text{vec}(\mathbf{X}^{gt}) \cdot \text{vec}(\mathbf{X}^{gt})^{\top}$. Note that $K_{pq}$ represents the element in the $p$-th row and $q$-th column of $\mathbf{K}$ between two graphs. This affinity loss decouples from solver effects, focusing solely on affinity quality, enhancing robustness and applicability to various solvers.

With its higher-order information, UM3C shows significantly greater robustness against noise, compared to GANN (Wang et al., 2020a), which centers only on node similarity and structural alignment.

## 5.2 UNSUPERVISED LEARNING USING PSEUDO LABELS

As previously mentioned, creating labels like $\mathbf{X}^{gt}$ demands significant time and effort. Hence, we generate pseudo labels $\mathbf{X}^{psd}$ using our learning-free solver M3C, aiming to replace $\mathbf{X}^{gt}$. To enhance the quality of $\mathbf{X}^{psd}$, we propose two techniques: affinity fusion and pseudo label selection.

The quality of $\mathbf{X}^{psd}$ hinges on input affinity $\mathbf{K}$. However, at the beginning of the training, the learned affinity $\mathbf{K}^{learn}$ is random and unreliable. Conversely, the hand-crafted affinity $\mathbf{K}^{raw}$ captures only geometric information of node pairs, i.e. distances and angles, limiting its expressiveness. Consequently, we fuse both affinities, leveraging their strengths.

In particular, for its simplicity and experimental effectiveness, we linearly merge learned and hand-crafted affinities, balanced by hyperparameter $\alpha$ as $\mathbf{K} = \mathbf{K}^{learn} + \alpha \mathbf{K}^{raw}$. The hand-crafted $\mathbf{K}^{raw}$ adheres to a standard procedure (Yan et al., 2015b; Jiang et al., 2021). Both affinities are normalized to the same scale. This design capitalizes on the reliability of hand-crafted affinity and the expressiveness of learned affinity. In initial epochs, $\mathbf{K}^{raw}$ enhances the quality of pseudo matching $\mathbf{X}^{psd}$. Later, learned affinity $\mathbf{K}^{learn}$ surpasses $\mathbf{K}^{raw}$, further refining the final result.

Furthermore, we optimize pseudo label pair selection for loss computation, guided by the relaxed indicator $\hat{\mathbf{C}}$. Unlike GANN, which accumulates losses for all pairs within inferred clusters, where a single incorrect assignment affects multiple pseudo label pairs of differing categories. UM3C adheres to $\hat{\mathbf{C}}$ and selects graph pairs with higher affinity rank as pseudo labels. This strategy, assuming higher affinity indicates greater accuracy, enhances pseudo affinity $\mathbf{K}^{psd}$ quality. The overall loss is:

$$\mathcal{L}_{all} = \sum_{ij} \hat{c}_{ij} \cdot \mathcal{L}(\mathbf{K}_{ij}^{learn}, \mathbf{K}_{ij}^{psd}). \qquad (9)$$

This approach chooses more accurate matching pairs, bringing pseudo affinity closer to ground truth. Empirical validation of this approach can be found in Sec. 6.3.

# 6 EXPERIMENTS

## 6.1 PROTOCOLS

Experiments on all learning-free solvers were conducted on a laptop with a 2.30GHz 4-core CPU and 16GB RAM using Matlab R2020a. All learning-based experiments were carried out on a Linux workstation with Xeon-3175X@3.10GHz CPU, one RX8000, and 128GB RAM.

**Datasets** We evaluate using two widely recognized datasets, Willow ObjectClass (Cho et al., 2013) and Pascal VOC (Everingham et al., 2010). Detailed introduction and implementation of the datasets will be introduced in Sec. G.1. For convenience of notation, $N_c$ and $N_g$ (denoted as $N_c \times N_g$) represent the number of categories and graphs we selected and mixed for tests on MGMC.

**Methods** We present three method versions: M3C-hard, M3C, and UM3C. M3C-hard serves as a baseline following the hard clustering MM framework (Sec. 4.1), employing Spectral Clustering and MGM-Floyd. M3C represents the relaxed algorithm from Sec. 4.3, and UM3C is the unsupervised learning model described in Sec. 5, both using the **fuse-rank** scheme, if not otherwise specified. We evaluate our methods in both learning-free and learning-based contexts. In learning-free experiments, we mainly compare M3C with DPMC (Wang et al., 2020b) and MGM-Floyd (Jiang et al., 2021), following protocols from Wang et al. (2020b;a). In learning-based experiments, we compare UM3C with unsupervised method GANN (Wang et al., 2020a), and supervised learning method BBGM (Rolínek et al., 2020), NGMv2 (Wang et al., 2020), GCAN (Jiang et al., 2022b), and COMMON (Lin et al., 2023).

Table 2: Evaluation of matching and clustering metric with inference time for the mixture graph matching and clustering on Willow Object Class. Following the previous work (Wang et al., 2020a), We select Car, Duck, and Motorbike as the cluster classes.

| Model | Learning | $N_c = 3, N_g = 8$, 0 outlier | | | | | $N_c = 3, N_g = 8$, 2 outliers | | | | | $N_c = 3, N_g = 8$, 4 outliers | | | | |
|---|---|---|---|---|---|---|---|---|---|---|---|---|---|---|---|---|
| | | MA ↑ | CA ↑ | CP ↑ | RI ↑ | time(s) ↓ | MA ↑ | CA ↑ | CP ↑ | RI ↑ | time(s) ↓ | MA ↑ | CA ↑ | CP ↑ | RI ↑ | time(s) ↓ |
| RRWM | free | 0.748 | 0.815 | 0.879 | 0.871 | **0.4** | 0.595 | 0.541 | 0.643 | 0.680 | **0.4** | 0.572 | 0.547 | 0.661 | 0.685 | **0.6** |
| MatchLift | free | 0.764 | 0.769 | 0.843 | 0.839 | 7.8 | 0.530 | 0.612 | 0.726 | 0.730 | 10.6 | 0.512 | 0.582 | 0.701 | 0.709 | 11.5 |
| MatchALS | free | 0.635 | 0.571 | 0.689 | 0.702 | 1.3 | 0.245 | 0.39 | 0.487 | 0.576 | 2.5 | 0.137 | 0.383 | 0.480 | 0.571 | 2.6 |
| CAO-C | free | 0.875 | 0.860 | 0.908 | 0.903 | 3.3 | **0.727** | 0.574 | 0.678 | 0.704 | 3.7 | **0.661** | 0.562 | 0.674 | 0.695 | 4.9 |
| MGM-Floyd | free | 0.879 | **0.931** | **0.958** | **0.952** | 2.0 | 0.716 | 0.564 | 0.667 | 0.696 | 2.3 | 0.653 | 0.580 | 0.690 | 0.708 | 2.9 |
| DPMC | free | 0.872 | 0.890 | 0.931 | 0.923 | 1.2 | 0.672 | 0.617 | 0.724 | 0.733 | 1.4 | 0.630 | 0.600 | 0.707 | 0.722 | 2.3 |
| M3C-hard | free | 0.838 | 0.855 | 0.907 | 0.899 | 0.4 | 0.620 | 0.576 | 0.684 | 0.705 | 0.6 | 0.596 | 0.587 | 0.694 | 0.713 | 0.7 |
| **M3C (ours)** | free | **0.884** | 0.911 | 0.941 | 0.938 | 0.5 | 0.687 | **0.653** | **0.750** | **0.758** | 0.6 | 0.635 | **0.646** | **0.748** | **0.753** | 1.0 |
| NGMv2 | sup. | 0.885 | 0.801 | 0.843 | 0.825 | 9.0 | 0.780 | 0.927 | 0.952 | 0.941 | 4.7 | 0.744 | 0.886 | 0.916 | 0.906 | 4.7 |
| BBGM | sup. | 0.939 | 0.704 | 0.751 | 0.758 | **1.6** | 0.806 | 0.964 | 0.977 | 0.971 | 4.8 | 0.747 | 0.881 | 0.918 | 0.908 | 6.6 |
| GANN | unsup. | 0.896 | 0.963 | 0.976 | 0.970 | 5.2 | 0.610 | 0.889 | 0.918 | 0.913 | 20.6 | 0.461 | 0.847 | 0.893 | 0.881 | 30.2 |
| **UM3C (ours)** | unsup. | **0.955** | **0.983** | **0.988** | **0.988** | 3.2 | **0.858** | **0.984** | **0.989** | **0.986** | **3.3** | **0.815** | **0.981** | **0.987** | **0.986** | **3.6** |

Table 3: Evaluation of matching and clustering metric with inference time for the mixture graph matching and clustering on Pascal VOC. Mixture classes are randomly picked and average values of metrics are reported over all the combinations. "**+UM3C(ours)**" means that UM3C apply the same feature extractor and two-graph solver with compared models. The pretrain weight of the feature extractor is loaded during the training to obtain a valid initial solution. Note that NGMv2 and BBGM are grouped together since their feature extractors are totally the same.

| Model | Learning | $N_c = 3, N_g = 8$ | | | | | $N_c = 5, N_g = 10$ | | | | |
|---|---|---|---|---|---|---|---|---|---|---|---|
| | | MA↑ | CA↑ | CP↑ | RI↑ | time(s)↓ | MA↑ | CA↑ | CP↑ | RI↑ | time(s)↓ |
| GANN | unsup. | 0.2774 | 0.6949 | 0.7613 | 0.768 | 33.785 | 0.2372 | 0.5103 | 0.599 | 0.7816 | 64.015 |
| UM3C | unsup. | **0.4979** | **0.7015** | **0.769** | **0.7756** | **5.2991** | **0.4817** | **0.5551** | **0.631** | **0.7921** | **24.661** |
| NGMv2 | sup. | **0.8114** | 0.755 | 0.8083 | 0.8165 | 4.2586 | **0.821** | 0.6087 | 0.689 | 0.8167 | 18.8087 |
| BBGM | sup. | 0.7919 | 0.7973 | 0.8406 | 0.8371 | **2.2618** | 0.7926 | 0.7261 | 0.783 | 0.8656 | **8.5146** |
| **+UM3C(ours)** | unsup. | 0.7928 | **0.8761** | **0.9065** | **0.9061** | 5.35 | 0.7862 | **0.7861** | **0.832** | **0.8989** | 24.8569 |
| GCAN | sup. | **0.8049** | 0.8089 | 0.8537 | 0.8438 | 4.6041 | **0.8041** | 0.672 | 0.7376 | 0.8459 | **23.8998** |
| **+UM3C(ours)** | unsup. | 0.75 | **0.824** | **0.8637** | **0.8565** | 7.5568 | 0.753 | **0.6846** | **0.7452** | **0.847** | 37.3579 |
| COMMON | sup. | 0.8334 | 0.9318 | 0.9467 | 0.9458 | **0.9152** | 0.8334 | 0.8058 | 0.848 | 0.9122 | **2.9551** |
| **+UM3C(ours)** | unsup. | **0.8435** | **0.9494** | **0.9629** | **0.9595** | 4.2193 | **0.8396** | **0.8129** | **0.8586** | **0.919** | 12.217 |

**Evaluation Metrics** We employ matching accuracy (MA), clustering purity (CP), rand index (RI), clustering accuracy (CA), and time cost as evaluation metrics, following prior research (Wang et al., 2020b;a). MA assesses matching performance, while CP, RI, CA represent the quality of cluster division. Detailed mathematical definitions are provided in Sec. G.2. Mean results from 50 tests are reported unless specified otherwise.

## 6.2 PERFORMANCE ON MGMC

We conduct mixture graph matching and clustering experiments on Pascal VOC and Willow Object-Class, as detailed in Table 2 and Table 3. For Pascal VOC, we explore two size settings: 3 clusters × 8 images and 5 clusters × 10 images, where clusters and images are randomly chosen from the dataset. In the Willow ObjectClass dataset, we use the 3 clusters × 8 images setting while investigating the impact of outliers. This setting follows the previous work Wang et al. (2020a), selecting Car, Duck, and Motorbike as the cluster classes, with images randomly sampled for each class.

Table 2 presents the performance of our learning-free solver, M3C, demonstrating its competitiveness compared to other learning-free algorithms. M3C achieves top matching accuracy (over 1% gain) in the settings without outliers. As the number of outliers increases, M3C's strength in clustering metrics becomes apparent, with gains of 3% - 5% in clustering accuracy. M3C also significantly outperforms M3C-hard, affirming the effectiveness of our designed relaxed indicator.

Our unsupervised model, UM3C, excels in both matching and clustering tasks. When compared to the peer method GANN, UM3C showcases superior performance on both Pascal VOC and Willow ObjectClass, achieving remarkable improvements of 5.9% - 24.45% in matching accuracy and 0.66% - 2% in clustering metrics. This advantage becomes more pronounced in the presence of outliers, underscoring the robustness of our unsupervised approach. On the other hand, UM3C remains competitive even when compared to supervised models. On the Willow ObjectClass dataset, UM3C outperforms supervised models (BBGM and NGMv2) in both matching and clustering metrics (1.6% - 7.8% in MA and 1.2% - 23% in CA, CP, RI). Concerning PascalVOC, a more challenging dataset, employing UM3C can further enhance the model's capabilities based on pretrained feature extractors, especially in clustering results.

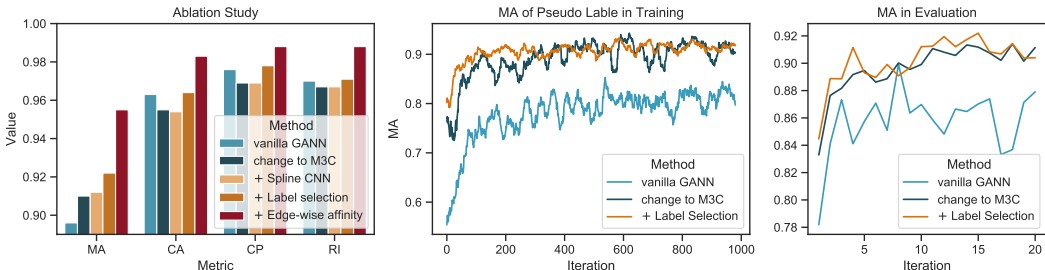

Figure 3: Ablation study of M3C by adding components under $3 \times 8$ Willow ObjectClass dataset without outliers. Left: Evaluation on matching and clustering performance. Right: Quality of pseudo labels during training and evaluation by iteration.

Furthermore, both M3C and UM3C stand out as time-efficient algorithms. As demonstrated in Table 2, M3C consumes only $1.25 - 1.67\times$ the time of the two-graph solver RRWM, while outpacing peer MGM methods by $2.3 - 11.5\times$ in terms of speed. On the other hand, UM3C requires only $0.54 - 2\times$ the time of the two-graph matching models NGMv2 and BBGM, delivering a speed advantage of $2.5 - 6.3\times$ over the peer method GANN. Moreover, the "+UM3C" extension incurs a time cost of only $1.2 - 2.9$s for the $3 \times 8$ setting and $6.0 - 14.5$s for the $5 \times 10$ setting, significantly shorter than the time required by GANN. It is important to note that the time-cost of UM3C in Table 3 exhibits significant variability, primarily due to different two-graph solvers and initialization methods.

## 6.3 Ablation Study

We evaluate UM3C on Willow ObjectClass with 3 clusters $\times$ 8 images, excluding outliers, to assess the effectiveness of different model components. We establish a baseline by substituting M3C for the GA-GM solver in GANN. Given that GANN lacks Spline CNN for feature refinement, we also investigate the impact of introducing Spline CNN in the unsupervised method. The effectiveness of label selection and edge-wise affinity learning is validated by adding each component successively.

The pseudo label selection results, measured by the matching accuracy of pseudo labels during both training and evaluation, are shown on the right side of Fig. 3. It shows a $5\%$ improvement over baseline M3C in the early training stage (first 100 iterations), affirming its ability to select pseudo labels closer to the ground truth.

Regarding edge-wise affinity learning, the left side of Fig. 3 illustrates its significant contribution to our model, highlighting the inadequacy of hand-crafted affinity and the necessity of learning edge-wise affinity. Spline CNN further enhances matching accuracy. Notably, our baseline method already outperforms GANN, attributed to our solver M3C.

## 6.4 Additional Experiments

We provide an index of additional experiments in the appendix for your reference. (H.1) Varying cluster number and cluster size: results on more diverse settings on Willow ObjectClass. (H.2) Comparison of different ranking schemes. (H.3) Comparison of different clustering algorithms. (H.4) Convergence study of M3C. (H.5) Hyperparameter study of M3C. (H.6) Generalization test of learned affinity $\mathbf{K}^{learn}$. (I) Visualization of matching results.

## 7 Conclusion and Outlook

We have presented a principled approach for jointly solving graph matching and clustering in scenarios involving mixed modes. Our learning-free solver, M3C, aligns with the minorize-maximization paradigm and introduces a relaxed cluster indicator to improve algorithm flexibility. Additionally, we integrate the M3C solver into an unsupervised learning pipeline, resulting in UM3C, with edge-wise affinity learning and pseudo label selection schemes. Remarkably, our methods outperform all state-of-the-art methods in the context of joint graph matching and clustering, which we believe is a practical setting to advance the research of graph matching. For future work, we may apply our methods to other domains like protein docking (Wu et al., 2024) and combinatorial problem instance generation (Li et al., 2023; Chen et al., 2024) whereby graph matching can be a building block.

## REPRODUCIBILITY STATEMENT

To foster reproducibility, we will make our code available at https://github.com/Thinklab-SJTU/M3C. We give details of our models M3C and UM3C in 'Implementation Details' Appendix. F, and our experimental protocol in 'Experiment Details' Appendix. G.

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

APPENDIX

## A  NOTATION

We first present all notations used in this paper for a better understanding of proposed algorithms and to facilitate the following discussion.

Table 4: Main notations and description used in this paper.

| Notations | Descriptions |
|---|---|
| $N$ | Number of input graphs. |
| $\mathbb{G}$ | $\mathbb{G}$ is a set of $N$ graphs $\mathbb{G} = \{\mathcal{G}_1 \ldots \mathcal{G}_N\}$. |
| $\mathcal{G}$ | Graph to be matched with vertex set $\mathcal{V}_\mathcal{G}$ and edge set $\mathcal{E}_\mathcal{G}$. |
| $\mathbb{X}$ | $\mathbb{X}$ denotes all the possible pairwise matching results in graph set $\mathbb{G}$: $\mathbb{X} = \{\mathbf{X}_{ij}\}_{1 \le i,j \le N}$. |
| $\mathbf{X}_{ij}(\mathbf{X}_{\mathcal{G}_i,\mathcal{G}_j})$ | $\mathbf{X}_{ij}(\mathbf{X}_{\mathcal{G}_i,\mathcal{G}_j}) \in \{0,1\}^{n_i \times n_j}$ denotes the pairwise matching results between $\mathcal{G}_i$ and $\mathcal{G}_j$. |
| $\mathbf{K}_{ij}$ | $\mathbf{K}_{ij} \in \mathbb{R}^{n_i n_j \times n_i n_j}$ denotes the affinity matrix between $\mathcal{G}_i$ and $\mathcal{G}_j$. Its diagonal and off-diagonal elements store the node-to-node and edge-to-edge affinities, respectively. |
| $J_{ij}$ | $J_{ij} = \text{vec}(\mathbf{X}_{ij})^\top \mathbf{K}_{ij} \text{vec}(\mathbf{X}_{ij})$ denotes the affinity score of graph pair $(\mathcal{G}_i, \mathcal{G}_j)$. |
| $\mathbf{C}$ | $\mathbf{C} \in \{0,1\}^{N \times N}$ is the cluster indicator of $N$ graphs. $c_{ij} = 1$ if $\mathcal{G}_i$ and $\mathcal{G}_j$ belongs to the same cluster, and $c_{ij} = 0$ otherwise. |
| $n, n_o$ | number of nodes in a graph and the number of its outliers |
| $N_c, N_g$ | $N_c$ denotes the number of clusters. The size of each cluster is represented as $\{N_{g_1}, N_{g_2}, \ldots, N_{g_{N_c}}\}$. When the size of clusters is the same, we shorthand it as $N_g$. |
| $\hat{\mathbf{C}}$ | $\hat{\mathbf{C}} \in \{0,1\}^{N \times N}$ is the relaxed cluster indicator by solving relaxed constraints. |
| $\mathbf{A}$ | $\mathbf{A} \in \{0,1\}^{N \times N}$ denotes the adjacency matrix for supergraph (Def. 2). |
| $\text{vec}(\cdot)$ | Column-vectorized operation of given matrix. |
| $\text{SCC}(\cdot)$ | The number of strong connection components of the input cluster. |
| $r$ | Hyperparameter for cluster relaxation. $r$ stands for the ratio of graph pairs to choose. |
| $\alpha$ | Hyperparameter for affinity construction. $\alpha$ is the weight of the hand-craft affinity. |

## B  COMPARISON WITH PREVIOUS WORKS

We underscore some key differences between our proposed method and two previous works, DPMC (Wang et al., 2020b) and GANN (Wang et al., 2020a; 2023), focusing on MGMC in Table 1. This comparison is to provide a comprehensive illustration of our novel contributions and substantial advancements in this field.

Some key points of differentiation are worth emphasizing:

- Both approaches employed rigid optimization structures (a tree and a fully connected graph), while M3C utilizes a discrete and partly connected supergraph, enhancing efficiency and flexibility in matching and clustering.
- DPMC lacks a convergence guarantee, and GANN requires hundreds of iterations for convergence, whereas M3C achieves convergence with fewer iterations.
- While GANN claimed to jointly optimize matching and clustering, empirical results reveal its inability to adapt clustering once it incorporates clustering during matching, which becomes a variant of hard clustering. In contrast, M3C demonstrates the crucial capability to jointly adjust its relaxed indicator alongside matching, a defining feature of a true joint optimization framework.
- GANN is rooted in Koopmans-Beckmann's QAP (Koopmans & Beckmann, 1957), which only considers structural similarity during matching, limiting its generalizability to edge feature learning. In contrast, UM3C utilizes a more generalized Lawler's QAP (Loiola et al., 2007) variation and introduces edge-wise affinity learning, making it more versatile.

## C CONVERGENCE ANALYSIS OF M3C

### C.1 CLUSTER DIVISION AND CLUSTER INDICATOR

In this section, we show the relationship between cluster division and cluster indication. We prove that the **transitive relation** $c_{ij}c_{jk} \leq c_{ik}$ is the sufficient and necessary condition for $\mathbf{C}$ to be a strict cluster division, which is proposed in Def. 3 of Sec. 3.

**Proposition C.1.** *Transitive relation $c_{ij}c_{jk} \leq c_{ik}$ is the sufficient and necessary condition for $\mathbf{C}$ to be a strict cluster division, where $\mathbf{C} \in \{0,1\}^{N \times N}$ and $c_{ij}$ denotes whether $\mathcal{G}_i$ and $\mathcal{G}_j$ are belong to the same category.*

*Proof.* **Necessary condition**: $\mathbf{C}$ is a strict cluster division $\Longrightarrow c_{ij}c_{jk} \leq c_{ik}$.

- When $c_{ij} = 1$ and $c_{jk} = 1$, $\mathcal{G}_i$ and $\mathcal{G}_j$ are of the same class, where $\mathcal{G}_j$ and $\mathcal{G}_k$ are of the same class, too. Therefore, $\mathcal{G}_i$ and $\mathcal{G}_k$ are of the same class, which means $c_{ik} = 1$ and $c_{ij}c_{jk} \leq c_{ik}$ holds.
- When one of $c_{ij}$ and $c_{jk}$ is equal to 1 and another equals 0, we assume $c_{ij} = 1$ and $c_{jk} = 0$ without loss of generality. That is, $\mathcal{G}_i$ and $\mathcal{G}_j$ are of the same class while $\mathcal{G}_j$ and $\mathcal{G}_k$ are not. Therefore, $\mathcal{G}_i$ and $\mathcal{G}_k$ are not of the same class either, which means $c_{ik} = 0$ and $c_{ij}c_{jk} = 0 \leq c_{ik}$ also holds.
- When $c_{ij} = 0$ and $c_{jk} = 0$, $\mathcal{G}_i$ and $\mathcal{G}_j$ are not of the same class, while $\mathcal{G}_j$ and $\mathcal{G}_k$ are neither of the same class. In that case, we cannot tell the relationship between $\mathcal{G}_i$ and $\mathcal{G}_k$, so $c_{ik} = 0/1$. It still holds that $c_{ij}c_{jk} \leq c_{ik}$.

**Sufficient condition**: $c_{ij}c_{jk} \leq c_{ik} \Longrightarrow \mathbf{C}$ is a strict cluster division.

- If $\mathcal{G}_i$ and $\mathcal{G}_j$, $\mathcal{G}_j$ and $\mathcal{G}_k$ are both of the same class, $c_{ij}c_{jk} = 1 \leq c_{ik}$, which means $c_{ik} = 1$ and $\mathcal{G}_i$ and $\mathcal{G}_k$ are of the same class.
- If $\mathcal{G}_i$ and $\mathcal{G}_j$ are of the same class, and $\mathcal{G}_j$ and $\mathcal{G}_k$ are of different classes, we have $c_{ij} = c_{ji} = 1$ and $c_{jk} = 0$. We find that $c_{ik} = c_{ji}c_{ik} \leq c_{jk} = 0$, thus, $c_{ik} = 0$ and $\mathcal{G}_i$ and $\mathcal{G}_k$ are not of the same class.

Above all, transitive relation $c_{ij}c_{jk} \leq c_{ik}$ is the sufficient and necessary condition for $\mathbf{C}$ to be a strict cluster division. $\qquad\square$

### C.2 PROOF OF THE CONVERGENCE OF MINORIZE-MAXIMIZATION FRAMEWORK

In this section, we prove the convergence of the MM Framework.

The objective function $\mathcal{F}(\mathbb{X}, \mathbf{C})$ is defined in Eq. 2 as:

$$
\begin{aligned}
\max_{\mathbb{X},\mathbf{C}} \mathcal{F}(\mathbb{X},\mathbf{C}) &= \max_{\mathbb{X},\mathbf{C}} \frac{\sum_{ij} c_{ij} \cdot \text{vec}(\mathbf{X}_{ij})^\top \mathbf{K}_{ij} \text{vec}(\mathbf{X}_{ij})}{\sum_{ij} c_{ij}} \\
s.t.\ \ &\mathbf{X}_{ij}\mathbf{1}_{n_j} \leq \mathbf{1}_{n_i}, \mathbf{X}_{ij}^\top \mathbf{1}_{n_i} \leq \mathbf{1}_{n_j}, \quad c_{ik}c_{kj} \leq c_{ij}, \forall i,j,k,\ \text{SCC}(\mathbf{C}) = N_c, \\
&\mathbf{X}_{ij} \in \{0,1\}^{n_i \times n_j}, c_{ij} \in \{0,1\}.
\end{aligned}
\tag{10}
$$

Let the $\mathbf{C} = h(\mathbb{X})$ solve the optimal cluster division for $\mathbb{X}$, and $f(\mathbb{X}) = \mathcal{F}(\mathbb{X}, h(\mathbb{X}))$ be an objective function with single variable $\mathbb{X}$. Without loss of generality, we have

$$
\max_{\mathbb{X}} f(\mathbb{X}) = \max_{\mathbb{X}} \mathcal{F}(\mathbb{X}, h(\mathbb{X})) = \max_{\mathbb{X}} \max_{\mathbf{C}|\mathbb{X}} \mathcal{F}(\mathbb{X},\mathbf{C}) = \max_{\mathbb{X},\mathbf{C}} \mathcal{F}(\mathbb{X},\mathbf{C})
\tag{11}
$$

Therefore, $f(\mathbb{X})$ is a variant of the objective function.

The MM algorithm works by finding a surrogate function that minorizes the objective function $f(\mathbb{X})$. Optimizing the surrogate function will either improve the value of the objective function or leave it unchanged. To optimize the surrogate function, two steps are conducted iteratively.

- **Construction.** The surrogate function $g(\mathbb{X}|\mathbb{X}^{(t)}) = \mathcal{F}(\mathbb{X}, h(\mathbb{X}^{(t)}))$ is constructed by inferring the best cluster based on current matching results $\mathbb{X}^{(t)}$:

$$
\begin{aligned}
h(\mathbb{X}^{(t)}) &= \arg\max_{\mathbf{C}} \sum_{ij} c_{ij} \cdot \text{vec}(\mathbf{X}_{ij}^{(t)})^\top \mathbf{K}_{ij}\text{vec}(\mathbf{X}_{ij}^{(t)}) \\
s.t.\ \ &c_{ik}c_{kj} \leq c_{ij}, \forall i,j,k,\ \text{SCC}(\mathbf{C}) = N_c.
\end{aligned}
\tag{12}
$$

- **Maximization.** Maximize $g(\mathbb{X}|\mathbb{X}^{(t)})$ instead of $f(\mathbb{X})$, which is solved by off-the-shelf graph matching solver.

$$\mathbb{X}^{(t+1)} = \arg\max_{\mathbb{X}} g(\mathbb{X}|\mathbb{X}^{(t)}) = \arg\max_{\mathbb{X}} f(\mathbb{X}, h(\mathbb{X}^{(t)}))$$

$$= \arg\max_{\mathbb{X}} \sum_{ij} c_{ij}^{(t)} \cdot \text{vec}(\mathbf{X}_{ij})^{\top} \mathbf{K}_{ij} \text{vec}(\mathbf{X}_{ij}) \tag{13}$$

$$s.t. \ \mathbf{X}_{ij}\mathbf{1}_{n_j} \leq \mathbf{1}_{n_i}, \mathbf{X}_{ij}^{\top}\mathbf{1}_{n_i} \leq \mathbf{1}_{n_j}.$$

The **Construction** step ensures two properties that $g(\mathbb{X}|\mathbb{X}^{(t)})$ holds: 1) Since $f(\mathbb{X}) = \mathcal{F}(\mathbb{X}, h(\mathbb{X}))$ adopts the optimal cluster division, it holds for all $\mathbb{X}$ that

$$g(\mathbb{X}|\mathbb{X}^{(t)}) = \mathcal{F}(\mathbb{X}, h(\mathbb{X}^{(t)})) \leq \mathcal{F}(\mathbb{X}, h(\mathbb{X})) = f(\mathbb{X}) \tag{14}$$

2) It also holds that

$$g(\mathbb{X}^{(t)}|\mathbb{X}^{(t)}) = \mathcal{F}(\mathbb{X}^{(t)}, h(\mathbb{X}^{(t)})) = f(\mathbb{X}^{(t)}). \tag{15}$$

With Eq. 14 and Eq. 15, for each iteration, the objective function will never decrease as shown in Eq. 3,

$$f(\mathbb{X}^{(t+1)}) \geq g(\mathbb{X}^{(t+1)}|\mathbb{X}^{(t)}) \geq g(\mathbb{X}^{(t)}|\mathbb{X}^{(t)}) = f(\mathbb{X}^{(t)}). \tag{16}$$

Thus we finish the proof of the convergence of our proposed MM Framework.

### C.3 QUICK CONVERGENCE OF HARD CLUSTERING

In this section, we show the proof of the proposition proposed in Sec. 4.2 that the hard clustering framework will converge quickly.

**Proposition C.2** (Proposition 4.1). *If the size of each cluster is fixed, the hard cluster indicator converges to the local optimum in one step:*

$$\mathbf{C}^{(t)} = \mathbf{C}^{(t+1)}, \ if \ \{N_{g_1}^{(t)}, N_{g_2}^{(t)}, \ldots, N_{g_{N_c}}^{(t)}\} = \{N_{g_1}^{(t+1)}, N_{g_2}^{(t+1)}, \ldots, N_{g_{N_c}}^{(t+1)}\}. \tag{17}$$

*Proof.* According to the maximization step in the MM framework, the pairwise matching is updated individually. The improvement on $\mathbf{X}_{ij}$ does not influence the optimization of other pairs. Therefore, it holds that,

$$\begin{aligned} J_{ij}^{(t+1)} &\geq J_{ij}^{(t)}, \ \forall c_{ij}^{(t)} = 1 \\ J_{ij}^{(t+1)} &= J_{ij}^{(t)}, \ \forall c_{ij}^{(t)} = 0 \end{aligned} \tag{18}$$

where $J_{ij} = \text{vec}(\mathbf{X}_{ij})^{\top}\mathbf{K}_{ij}\text{vec}(\mathbf{X}_{ij})$ denotes the pairwise affinity score.

We prove the proposition by contradiction. Assume that $\mathbf{C}^{(t)} \neq \mathbf{C}^{(t+1)}$. According to the optimization framework, we have that

$$f(\mathbb{X}^{(t+1)}) \geq g(\mathbb{X}^{(t+1)}|\mathbb{X}^{(t)}) \geq g(\mathbb{X}^{(t)}|\mathbb{X}^{(t)}), \tag{19}$$

which means,

$$\mathcal{F}(\mathbb{X}^{(t+1)}, \mathbf{C}^{(t+1)}) \geq \mathcal{F}(\mathbb{X}^{(t+1)}, \mathbf{C}^{(t)}) \geq \mathcal{F}(\mathbb{X}^{(t)}, \mathbf{C}^{(t)}). \tag{20}$$

Therefore,

$$\begin{aligned} &\mathcal{F}(\mathbb{X}^{(t+1)}, \mathbf{C}^{(t+1)}) - \mathcal{F}(\mathbb{X}^{(t+1)}, \mathbf{C}^{(t)}) \\ =&\frac{1}{\sum c_{ij}^{(t+1)}} \sum_{ij} c_{ij}^{(t+1)} \cdot J_{ij}^{(t+1)} - \frac{1}{\sum c_{ij}^{(t)}} \sum_{ij} c_{ij}^{(t)} \cdot J_{ij}^{(t+1)} \\ \geq&0. \end{aligned} \tag{21}$$

Since

$$\{N_{g_1}^{(t)}, N_{g_2}^{(t)}, \ldots, N_{g_{N_c}}^{(t)}\} = \{N_{g_1}^{(t+1)}, N_{g_2}^{(t+1)}, \ldots, N_{g_{N_c}}^{(t+1)}\}, \tag{22}$$

it holds that,

$$\sum c_{ij}^{(t+1)} = \sum c_{ij}^{(t)}. \tag{23}$$

---

**Algorithm 1:** Learning-free solver M3C

---

**Input:** Iterations number $T$, Cluster number $N_c$, Graph set $\{\mathcal{G}_1, \ldots \mathcal{G}_N\}$.

1 Construct affinity matrix $\mathbf{K}$ from node coordinates of graph set $\{\mathcal{G}_1, \ldots \mathcal{G}_N\}$ through standard process (Yan et al., 2015b; Jiang et al., 2021).

2 Obtain initialization matching $\mathbb{X}^{(0)}$ via a two-graph solver e.g. RRWM (Cho et al., 2010);

3 **for** $t = 1 : T$ **do**

    /\* Construction-Step \*/

4     Construct the surrogate function $g(\mathbb{X}|\mathbb{X}^{(t-1)})$ via solving $\hat{\mathbf{C}}^t = \hat{h}(\mathbb{X}^{(t-1)})$ in Eq. 6 with three strategy candidates: global-rank, local-rank, or fuse-rank;

    /\* Maximization-Step \*/

5     Set $\mathbf{A}^{(t)}$ as relaxed indicator $\hat{\mathbf{C}}^{(t)}$, maximizing $g(\mathbb{X}|\mathbb{X}^{(t-1)})$ via Eq. 7 to find optimal matching composition $\mathbb{X}$ on $\mathbf{A}^{(t)}$;

6 **end**

7 Obtain the affinity score for each graph pair: $J_{ij} = \text{vec}(\mathbf{X}_{ij})^\top \mathbf{K}_{ij} \text{vec}(\mathbf{X}_{ij})$;

8 Sparsification on the affinity score: $\{J_{ij}^k\} = \text{KNN}(\{J_{ij}\}, k)$;

9 Apply clustering algorithm, e.g. Spectral Clustering (Ng et al., 2002), Multi-Cut (Swoboda & Andres, 2017), on $\{J_{ij}^k\}$ to get $\mathbf{C}$.

**Output:** Matching $\mathbb{X}^{(t)}$, cluster $\mathbf{C}$.

---

According to Eq. 18, we can further have

$$
\mathcal{F}(\mathbb{X}^{(t+1)}, \mathbf{C}^{(t+1)}) - \mathcal{F}(\mathbb{X}^{(t+1)}, \mathbf{C}^{(t)})
$$

$$
= \frac{1}{\sum c_{ij}^{(t)}} \left( \sum_{ij} c_{ij}^{(t+1)} \cdot J_{ij}^{(t+1)} - \sum_{ij} c_{ij}^{(t)} \cdot J_{ij}^{(t+1)} \right)
$$

$$
= \frac{1}{\sum c_{ij}^{(t)}} \left( \sum_{c_{ij}^{(t+1)}=1, c_{ij}^{(t)}=0} J_{ij}^{(t+1)} - \sum_{c_{ij}^{(t+1)}=0, c_{ij}^{(t)}=1} J_{ij}^{(t+1)} \right) \tag{24}
$$

$$
= \frac{1}{\sum c_{ij}^{(t)}} \left( \sum_{c_{ij}^{(t+1)}=1, c_{ij}^{(t)}=0} J_{ij}^{(t)} - \sum_{c_{ij}^{(t+1)}=0, c_{ij}^{(t)}=1} J_{ij}^{(t+1)} \right)
$$

$$
\geq 0.
$$

Moreover, according to $J_{ij}^{(t+1)} \geq J_{ij}^{(t)}, \forall c_{ij}^t = 1$, we have

$$
\sum_{c_{ij}^{(t+1)}=1, c_{ij}^{(t)}=0} J_{ij}^{(t)} - \sum_{c_{ij}^{(t+1)}=0, c_{ij}^{(t)}=1} J_{ij}^{(t)} \geq \sum_{c_{ij}^{(t+1)}=1, c_{ij}^{(t)}=0} J_{ij}^{(t)} - \sum_{c_{ij}^{(t+1)}=0, c_{ij}^{(t)}=1} J_{ij}^{(t+1)}, \tag{25}
$$

which means

$$
\frac{1}{\sum c_{ij}^{(t+1)}} \sum_{ij} c_{ij}^{(t+1)} \cdot J_{ij}^{(t)} - \frac{1}{\sum c_{ij}^{(t)}} \sum_{ij} c_{ij}^{(t)} \cdot J_{ij}^{(t)} \geq 0. \tag{26}
$$

That is to say

$$
\mathcal{F}(\mathbb{X}^{(t)}, \mathbf{C}^{(t+1)}) \geq \mathcal{F}(\mathbb{X}^{(t)}, \mathbf{C}^{(t)}), \tag{27}
$$

which means $\mathbf{C}^{(t)}$ is not the optimal cluster division for $\mathbf{X}^{(t)}$. Contradiction. Therefore, we have $\mathbf{C}^{(t)} = \mathbf{C}^{(t+1)}$. $\qquad\qquad\square$

## D   Detailed Algorithm of M3C

Please refer to Algorithm 1 and Algorithm 2 for a detailed presentation of our proposed learning-free solver M3C and unsupervised learning method UM3C.

---

**Algorithm 2:** Unsupervised learning UM3C

---

**Input:** Images $\{\mathcal{I}_1, \ldots, \mathcal{I}_N\}$, node coordinates $\{\mathcal{V}_1, \ldots, \mathcal{V}_N\}$

1   Obtain node and edge features $\mathbf{F}^n, \mathbf{F}^e$ via VGG16 and SplineCNN;

2   Construct $\mathbf{K}^{learn}$ from $\mathbf{F}^n, \mathbf{F}^e$ by $\mathbf{K}_u^{learn} = (\mathbf{F}^n)^\top \mathbf{\Lambda} \mathbf{F}^n, \mathbf{K}_q^{learn} = (\mathbf{F}^e)^\top \mathbf{\Lambda} \mathbf{F}^e$;

3   Obtain hand-crafted $\mathbf{K}^{raw}$ following Zhou & Torre (2016); Yan et al. (2015b); Jiang et al. (2021); Wang et al. (2020b);

4   $\mathbf{K} = \mathbf{K}^{learn} + \alpha \mathbf{K}^{raw}$;

5   **if** *training* **then**

6      $\mathbf{X}^{psd}, \hat{\mathbf{C}} = \text{M3C}(\mathbf{K})$;

7      $\mathbf{K}^{psd} = \text{vec}(\mathbf{X}^{psd}) \cdot \text{vec}(\mathbf{X}^{psd})$;

8      $\mathcal{L}_{all} = \sum_{ij} \hat{c}_{ij} \cdot \mathcal{L}(\mathbf{K}_{ij}^{learn}, \mathbf{K}_{ij}^{psd})$

9   **else**

10      $\mathbf{X}, \mathbf{C} = \text{M3C}(\mathbf{K})$;

11   **end**

**Output:** Matching $\mathbb{X}$, cluster $\mathbf{C}$.

---

Table 5: Time complexity comparison of peer methods, where $N$ and $n$ is the total number of graphs and number of nodes for each graph (we assume equal size here for notation simplicity), $T_i$ denotes the iterations needed for the algorithm to converge, and $\tau_{pair}$ denotes the time cost of calling a two-graph matching solver.

| method | time complexity for MGMC |
|---|---|
| MGM-Floyd (Jiang et al., 2021) | $\mathcal{O}(N^4 n + N^3 n^3 + N^2 \tau_{pair} + t N N_c d)$ |
| CAO-C (Yan et al., 2015b) | $\mathcal{O}(N^4 n + N^3 n^3 + N^2 \tau_{pair} + t N N_c d)$ |
| DPMC (Wang et al., 2020b) | $\mathcal{O}(T_1 * (N^2 n^4 + N^2 \log N + t N N_c d) + N^2 \tau_{pair})$ |
| GA-MGMC (Wang et al., 2020a) | $\mathcal{O}(T_2 * (T' N^2 n^2 d + t N N_c d))$ |
| M3C (ours) | $\mathcal{O}(T_3 * (N^3 n^3 + N^2 \log N^2 + t N N_c d) + N^2 \tau_{pair})$ |

# E   TIME COMPLEXITY ANALYSIS BETWEEN LEARNING-FREE SOLVERS

We analyze the time complexity of our method M3C and compare it with other learning-free solvers in Table 5. Notably, our algorithm outperforms vanilla MGM-Floyd (Jiang et al., 2021) in terms of speedup due to a lower time complexity bound. Additionally, our approach benefits from a significantly reduced constant factor.

Let $N$ and $n$ denote the number of graphs and nodes in one graph (ignoring the different graph sizes for the brevity of notation), respectively, and $\tau_{pair}$ denote the time cost of a two-graph matching solver, such as RRWM (Cho et al., 2010). It costs $\mathcal{O}(N^2 \tau_{pair})$ to calculate the score matrix and $\mathcal{O}(N^2 \log N^2)$ to construct the supergraph. In the worst case, we would add $\frac{N(N-1)}{2}$ edges to the supergraph, resulting in $\mathcal{O}(N^3 n^3)$ time cost for performing MGM-Floyd. Additionally, spectral clustering is applied, with a time cost of $\mathcal{O}(t N N_c d)$, where $t$ is the k-means iteration, and $d$ is the dimension for embedded features. $T$ denotes the number of iterations that M3C takes to converge. Therefore, the total time complexity is given by $\mathcal{O}(T * (N^3 n^3 + N^2 \log N^2 + t N N_c d) + N^2 \tau_{pair})$.

It's important to note that M3C runs faster than peer methods in Table 2. This is primarily due to two reasons. Firstly, M3C exhibits a significantly lower constant factor in the maximization step, and the expected number of edges added is much smaller than in the worst case scenario. In practice, the actual run time of the maximization step can be approximated as $\theta N^3 n^3$, where $\theta < 1$. Empirical studies show that $\theta \approx 0.16$ for $N_c = 5$ and $N_g = 20$, and this value may further decrease with larger $N_c$ and $N_g$. Secondly, owing to the relaxed cluster indicator, M3C converges much faster than GA-GAGM, resulting in a significantly reduced number of iterations: $T_3 \ll T_2$. As a result, the actual running time of M3C is significantly less than that of DPMC and GA-MGMC.

## F IMPLEMENTATION DETAILS

In this section, we introduce more implementation details of M3C and UM3C, including the detailed structure of the neural network we applied, the construction of affinity matrix $\mathbf{K}$, and some hyper-parameters setting.

### F.1 NETWORK STRUCTURE FOR FEATURE EXTRACTION

We utilize the feature extractor described in Rolínek et al. (2020) with a few modifications. The process is outlined below:

- Compute the outputs of `relu4_2`, `relu5_1` of the VGG16 (Simonyan & Zisserman, 2014) network pre-trained on ImageNet (Krizhevsky et al., 2012), to obtain feature $\mathbf{F}_1$ and $\mathbf{F}_2$, respectively. These features are then concatenated to create the final CNN feature $\mathbf{F}$:

$$\mathbf{F} = \text{CONCAT}(\mathbf{F}_1, \mathbf{F}_2) \tag{28}$$

  The detailed network structure and parameters are shown in Table. 6.

- Feed the obtained feature $\mathbf{F}$ and the graph adjacency $\mathcal{A}$ into the geometric feature refinement component. The graph adjacency $\mathcal{A}$ is generated using Delaunay triangulation (Delaunay et al., 1934) based on keypoint locations. We apply SplineConv (Fey et al., 2018) to encode higher-order information and the geometric structure of the entire graph into node-wise features $\mathbf{F}^n$:

$$\mathbf{F}^n = \text{SplineConv}(\mathbf{F}, \mathcal{A}) \tag{29}$$

The Spline Conv operation is calculated as follows:

$$\mathbf{F}_i^n = \frac{1}{|\mathcal{N}(i)|} \sum_{j \in \mathcal{N}(i)} \mathbf{F}_j^n \cdot h_\Theta(\mathbf{E}_{i,j}) \tag{30}$$

where $\mathbf{F}_i^n$ represents the node feature of $v_i$, $\mathcal{N}(i)$ denotes the neighborhood of $v_i$, $\mathbf{E}_{i,j} = \mathbf{F}_i^n - \mathbf{F}_j^n$ stands for the edge feature of the edge between $v_i$ and $v_j$, and $h_\Theta$ denotes a kernel function defined over the weighted B-Spline tensor product basis.

### F.2 CONSTRUCTION OF AFFINITY MATRIX $\mathbf{K}$

In our paper, we discuss two types of affinities: the traditional $\mathbf{K}^{raw}$ and the learned $\mathbf{K}^{learn}$. The former is employed in all learning-free experiments, while both are utilized in the UM3C process.

Table 6: Network structure of vgg16_bn as applied in UM3C. The pre-trained weight is downloaded by PyTorch. The bold line denotes the `relu4_2`, `relu5_1`, and `final` layers whose outputs are applied as node features, edge features, and global features.

| | Layer | Channels | Kernel | | Layer | Channels | Kernel |
|---|---|---|---|---|---|---|---|
| | Conv2d | (3, 64) | (3, 3) | 3 | ReLU | (256, 256) | - |
| | BatchNorm2d | (64, 64) | - | | MaxPool2d | - | 2 |
| | ReLU | (64, 64) | - | | Conv2d | (256, 512) | (3, 3) |
| 1 | Conv2d | (64, 64) | (3, 3) | | BatchNorm2d | (512, 512) | - |
| | BatchNorm2d | (64, 64) | - | | ReLU | (512, 512) | - |
| | ReLU | (64, 64) | - | | **Conv2d** | **(512, 512)** | **(3, 3)** |
| | MaxPool2d | - | 2 | 4 | BatchNorm2d | (512, 512) | - |
| | Conv2d | (64, 128) | (3, 3) | | ReLU | (512, 512) | - |
| | BatchNorm2d | (128, 128) | - | | Conv2d | (512, 512) | (3, 3) |
| | ReLU | (128, 128) | - | | BatchNorm2d | (512, 512) | - |
| 2 | Conv2d | (128, 128) | (3, 3) | | ReLU | (512, 512) | - |
| | BatchNorm2d | (128, 128) | - | | MaxPool2d | - | 2 |
| | ReLU | (128, 128) | - | | **Conv2d** | **(512, 512)** | **(3, 3)** |
| | MaxPool2d | - | 2 | | BatchNorm2d | (512, 512) | - |
| | Conv2d | (128, 256) | (3, 3) | | ReLU | (512, 512) | - |
| | BatchNorm2d | (256, 256) | - | | Conv2d | (512, 512) | (3, 3) |
| | ReLU | (256, 256) | - | 5 | BatchNorm2d | (512, 512) | - |
| 3 | Conv2d | (256, 256) | (3, 3) | | ReLU | (512, 512) | - |
| | BatchNorm2d | (256, 256) | - | | Conv2d | (512, 512) | (3, 3) |
| | ReLU | (256, 256) | - | | BatchNorm2d | (512, 512) | - |
| | Conv2d | (256, 256) | (3, 3) | | ReLU | (512, 512) | - |
| | BatchNorm2d | (256, 256) | - | | **MaxPool2d** | **-** | **2** |

Table 7: Parameters of UM3C.

| param | WillowObject | | PascalVOC | | description |
|---|---|---|---|---|---|
| | $3 \times 8$ | $5 \times 10$ | $3 \times 8$ | $5 \times 10$ | |
| lr | $10^{-3}$ | $10^{-3}$ | $10^{-3}$ / $10^{-5}$ | $10^{-3}$ / $10^{-5}$ | learning rate |
| lr-steps | $\{100, 500\}$ | $\{100, 500\}$ | $\{100, 500\}$ | $\{100, 500\}$ | lr/=10 at these number of iterations |
| $\alpha$ | train: 1 
 test: 0 | train: 1 
 test: 1 | train: 0.5 
 test: 0.5 | train: 0.5 
 test: 0.5 | weight of $\mathbf{K}^{raw}$ |
| $\beta$ | 0.9 | 0.9 | 0.9 | 0.9 | weight parameter in $\mathbf{K}^{raw}$ |
| $\sigma^2$ | 0.03 | 0.03 | 0.03 | 0.03 | the scaling factor in $\mathbf{K}^{raw}$ |
| $T$ | 2 | 2 | 2 | 2 | max iterations of M3C |

The construction of $\mathbf{K}^{raw}$ adheres to the standard protocol used in previous works (Zhou & Torre, 2016; Yan et al., 2015a;b; Jiang et al., 2021; Wang et al., 2020b). $\mathbf{K}^{raw}$ has no node affinity and relies on edge affinity, which is computed based on two factors: length similarity and angle similarity. For each edge $e = ((x_1, y_1), (x_2, y_2))$, its length feature $d_e$ is calculated as $d_e = \sqrt{(x_1 - x_2)^2 + (y_1 - y_2)^2}$, and its angle feature $\theta_e$ is computed as $\theta_e = \tan^{-1}(\frac{y_1 - y_2}{x_1 - x_2 + \epsilon})$. The edge affinity is determined using the following formula:

$$k^{raw}e_1, e_2 = \exp\left(-\frac{1}{\sigma^2}\left(\beta\,|d_{e_1} - d_{e_2}| + (1 - \beta)\,|\theta_{e_1} - \theta_{e_2}|\right)\right) \tag{31}$$

The learned affinity $\mathbf{K}^{learn}$ is extracted using a deep learning model, following the standard pipeline (Rolínek et al., 2020; Wang et al., 2020a). Node features $\mathbf{F}_i^n$ are obtained through VGG16 and SplineConv, while edge features are constructed as:

$$\mathbf{F}_{ij}^e = \mathbf{F}_i^n - \mathbf{F}_j^n \tag{32}$$

The learned affinity matrix $\mathbf{K}$ is computed as:

$$\mathbf{K}_u = (\mathbf{F}^n)^\top \mathbf{\Lambda} \mathbf{F}^n, \quad \mathbf{K}_q = (\mathbf{F}^e)^\top \mathbf{\Lambda} \mathbf{F}^e \tag{33}$$

Here, $\mathbf{K}_u$ represents unary affinity, $\mathbf{K}_q$ denotes quadratic affinity, and $\mathbf{\Lambda}$ is set to $\mathbf{I}$ for stable training. UM3C constructs the affinity matrix through a combination:

$$\mathbf{K} = \mathbf{K}^{learn} + \alpha \mathbf{K}^{raw}, \tag{34}$$

where $\alpha$ is used to adjust the weight of $\mathbf{K}^{raw}$. Further parameter details are provided in Sec. F.3.

### F.3  PARAMETER SETTINGS

The detailed configuration of our model parameters is listed in Table 7, which are tuned based on their performance on the training data. The parameter $\beta$ and $\sigma$ for $\mathbf{K}^{raw}$ follows the parameter used for traditional solvers. The max iteration $T$ is chosen based on the performance, convergence, and time cost of M3C. For $\alpha$, we found that as $\mathbf{K}^{learn}$ gets better when the training proceeds, a less desirable $\mathbf{K}^{raw}$ would harm the performance of the solver under the simpler setting where $N_c = 3, N_g = 8$, but is still instructive under more complex setting where $N_c = 5, N_g = 10$. This also means that given more training categories, there is still room for improvement in unsupervised learning methods.

## G  EXPERIMENT DETAILS

### G.1  DATASETS

We conducted experiments on two datasets: Willow Object Class (Cho et al., 2013) and Pascal VOC Keypoint (Everingham et al., 2010).

The Willow Object Class dataset comprises 304 images gathered from Caltech-256 (Griffin et al., 2007) and Pascal VOC 2007 (Everingham et al., 2007). These images span 5 categories: 208 faces, 50 ducks, 66 wine bottles, 40 cars, and 40 motorbikes. Each image is annotated with 10 keypoints, and we introduced random outliers for robustness tests.

The Pascal VOC Keypoint dataset features natural images from 20 classes in VOC 2011 (Everingham et al., 2010), with additional keypoint labels provided by Bourdev & Malik (2009). To tailor the dataset to the graph matching and clustering problem, we selected 10 classes: aeroplane, bicycle, bird, cat, chair, cow, dog, horse, motorbike, and sheep. We filtered out images with incomplete keypoint counts, ensuring that all remaining images had 9-10 common keypoints for each class. We also added random outliers to ensure that all images consistently contained exactly 10 keypoints. This resulted in a training set of 944 images and an evaluation set of 220 images.

In both datasets, we constructed graphs using Delaunay triangulation. For learning-based models, images were cropped to object bounding boxes and resized to $256 \times 256$ pixels.

### G.2 EVALUATION METRIC

Denote a cluster with a set of graphs $\mathcal{C} = \{\mathcal{G}_1 \dots \mathcal{G}_n\}$. The ground truth cluster division is denoted as $\mathcal{C}^{gt}$, and the predicted cluster is denoted as $\mathcal{C}$. Moreover, let $\mathbf{C}^{gt}(c_{ij}^{gt})$ denotes the ground truth cluster indicator and $\mathbf{C}(c_{ij})$ denotes the predict cluster indicator. Performance metrics include both matching accuracy and clustering quality:

**Matching Accuracy (MA)**  We only consider the intra-cluster matching accuracy and thus by adapting the accuracy for a single cluster, we have

$$\text{MA} = \frac{1}{\sum c_{ij}^{gt}} \sum_{ij} c_{ij}^{gt} \cdot \text{ACC}(\mathbf{X}_{ij}), \tag{35}$$

where $\text{ACC}(\mathbf{X}_{ij})$ denotes accuracy for matching $\mathbf{X}_{ij}$. Here $\mathbf{C}$ refers to an indicator for strict cluster division.

**Clustering Purity (CP)**  (Manning et al., 2008): it is given by

$$\text{CP} = \frac{1}{N} \sum_{i=1}^{N_c} \max_{j \in \{1, \dots, N_c\}} \left| \mathcal{C}_i \cap \mathcal{C}_j^{gt} \right|, \tag{36}$$

where $\mathcal{C}_i'$ is the predicted cluster $i$ and $\mathcal{C}_j$ is the ground truth cluster $j$, and $N$ is the total number of graphs.

**Rand Index (RI)**  (Rand, 1971): RI calculates the correct graph pairs overall.

$$\text{RI} = \frac{1}{N^2} \cdot \left( \sum_{c_{ij}=1, c_{ij}^{gt}=1} 1 + \sum_{c_{ij}=0, c_{ij}^{gt}=0} 1 \right) \tag{37}$$

where $\sum_{c_{ij}=1, c_{ij}^{gt}=1} 1$ represents the number of graphs predicted in the same cluster with same label, $\sum_{c_{ij}=0, c_{ij}^{gt}=0} 1$ the number of pairs that are in different clusters with different labels, and it is normalized by the total number of graph pairs $N^2$.

**Clustering Accuracy (CA)**  (Wang et al., 2020b), it is defined by:

$$\text{CA} = 1 - \frac{1}{N_c} \left( \sum_{\mathcal{C}_a} \sum_{\mathcal{C}_a' \neq \mathcal{C}_b'} \frac{|\mathcal{C}_a' \cap \mathcal{C}_a| \, |\mathcal{C}_b' \cap \mathcal{C}_a|}{|\mathcal{C}_a| \, |\mathcal{C}_a|} + \sum_{\mathcal{C}_a'} \sum_{\mathcal{C}_a \neq \mathcal{C}_b} \frac{|\mathcal{C}_a' \cap \mathcal{C}_a| \, |\mathcal{C}_a' \cap \mathcal{C}_b|}{|\mathcal{C}_a| \, |\mathcal{C}_b|} \right) \tag{38}$$

where $\mathcal{C}_a, \mathcal{C}_b$ are the ground truth clusters and $\mathcal{C}_a', \mathcal{C}_b'$ denotes prediction.

### G.3 APPLY 2GM AND MGM ON MGMC.

To apply 2GM and MGM solvers on MGMC, we need to first get the pairwise matching results $\mathbb{X}$ from these solvers. Then we will generate respective clustering results $\mathbf{C}$ based on resulting matching $\mathbb{X}$ and given affinity matrix $\{\mathbf{K}_{ij}\}$. The details of the clustering algorithm are introduced in Sec. G.4. Both $\mathbb{X}$ and $\mathbf{C}$ are used for the evaluation of these solvers.

---

**Algorithm 3:** Clustering algorithm.

---

**Input:** Matching results $\mathbb{X}$, affinity matrix $\{\mathbf{K}_{ij}\}$.

1 Obtain the affinity score for each graph pair: $J_{ij} = \text{vec}(\mathbf{X}_{ij})^\top \mathbf{K}_{ij}\text{vec}(\mathbf{X}_{ij})$;

2 Sparsification on the affinity score: $\{J_{ij}^k\} = \text{KNN}(\{J_{ij}\}, k)$;

3 Apply clustering algorithm, e.g. Spectral Clustering (Ng et al., 2002), Multi-Cut (Swoboda & Andres, 2017), on $\{J_{ij}^k\}$ to get $\mathbf{C}$.

**Output:** Cluster $\mathbf{C}$.

---

### G.4 DETAILS FOR CLUSTERING ALGORITHM.

For all solvers (2GM, MGM, and MGMC), we adhere to the same clustering procedure outlined in Alg. 3. The first step involves computing the affinity score $J_{ij}$ for each pair of graphs. To enhance the effectiveness of clustering, we employ a sparsification technique, consistent with the pre-processing approach in Wang et al. (2020b), aimed at obtaining a more efficient input matrix. Specifically, when dealing with a pair of two graphs, if one graph is not among the $k$-nearest neighbors of the other, we set their corresponding scores $J_{ij}$ (and $J_{ji}$) to zero. The parameter $k$ is consistently set to 10 for all tests. The resulting sparsified affinity score is denoted as $J_{ij}^k$.

## H ADDITIONAL EXPERIMENTS

### H.1 VARYING CLUSTER NUMBER AND CLUSTER SIZE

We assess the model's generalization ability concerning the number of graphs and clusters. For $N_c = 3$, the categories consist of car, motorbike, and wine bottle. For $N_c = 4$, additional categories include face. We also investigate unbalanced cluster sizes for $N_c = 3$, comprising 20 cars, 10 motorbikes, and 5 wine bottles. Both GANN and UM3C are trained with $N_c = 5$ and $N_g = 10$, excluding outliers. During testing, two outliers are randomly added to the graph in all settings.

Table 8 demonstrates the robustness of our methods with varying cluster and graph numbers. Our learning-free solver, M3C, exhibits competitive performance compared to DPMC, with a matching accuracy ranging from 1% loss to 2% gain, and clustering accuracy improvement ranging from 2% to 9%. These achievements further reflect the superiority of our proposed ranking scheme.

Notably, our UM3C outperforms in all performance metrics. It consistently achieves a cluster accuracy above 0.97 and a matching accuracy exceeding 0.87 across all settings, representing a $1 \sim 7\%$ improvement in matching accuracy and $2 \sim 18\%$ enhancement in clustering accuracy. The fact that UM3C's training setting differs from testing settings validates the strong generalization ability of our method across varying numbers of graphs, clusters, and the presence of outliers. Furthermore, our method, although trained under simpler conditions, can be effectively deployed in more complex scenarios, yielding satisfactory performance.

### H.2 COMPARISON OF DIFFERENT RANKING SCHEMES

We compare three proposed ranking schemes across various test settings. M3C-Global, M3C-Local, and M3C-Fuse refer to global-rank, local-rank, and fuse-rank, respectively. We vary the number of graphs $N_g$ within each cluster while keeping the cluster number fixed at $N_c = 5$, and we also vary the number of outliers in each graph while maintaining $N_c = 5$ and $N_g = 20$.

Table 8: Evaluation of matching and clustering accuracy by varying the number of clusters, and number of graphs in each cluster on WillowObj. MA and CA are used for matching accuracy and clustering accuracy, respectively.

| $N_c \times N_g$ Metrics | Learning | $3 \times 20$, 2 outliers MA ↑ | CA ↑ | $4 \times 20$, 2 outliers MA ↑ | CA ↑ | $5 \times 20$, 2 outliers MA ↑ | CA ↑ | $5 \times 15$, 2 outliers MA ↑ | CA ↑ | $5 \times 10$, 2 outliers MA ↑ | CA ↑ | $3 \times [20,10,5]$, 2 outliers MA ↑ | CA ↑ |
|---|---|---|---|---|---|---|---|---|---|---|---|---|---|
| RRWM | free | 0.658 | 0.932 | 0.642 | 0.858 | 0.665 | 0.790 | 0.648 | 0.693 | 0.664 | 0.679 | 0.633 | 0.681 |
| CAO-C | free | 0.849 | 0.946 | 0.812 | 0.855 | 0.820 | 0.790 | 0.801 | 0.708 | 0.804 | 0.679 | 0.787 | 0.757 |
| MGM-Floyd | free | 0.845 | 0.945 | 0.812 | 0.878 | 0.819 | 0.807 | 0.798 | 0.727 | 0.799 | 0.707 | 0.778 | 0.755 |
| DPMC | free | **0.867** | 0.942 | 0.827 | 0.894 | 0.775 | 0.772 | 0.739 | 0.713 | 0.756 | 0.744 | **0.795** | 0.823 |
| M3C-hard | free | 0.758 | **0.966** | 0.782 | 0.908 | 0.726 | 0.824 | 0.710 | 0.753 | 0.722 | 0.719 | 0.727 | 0.744 |
| **M3C (ours)** | free | 0.857 | 0.961 | **0.851** | **0.933** | **0.835** | **0.900** | **0.812** | **0.805** | **0.809** | **0.780** | 0.792 | **0.881** |
| GANN | unsup. | 0.532 | 0.834 | 0.589 | 0.801 | 0.528 | 0.784 | 0.551 | 0.802 | 0.552 | 0.827 | 0.475 | 0.802 |
| **UM3C (ours)** | unsup. | **0.874** | **0.992** | **0.897** | **0.981** | **0.879** | **0.972** | **0.876** | **0.975** | **0.878** | **0.975** | **0.872** | **0.984** |

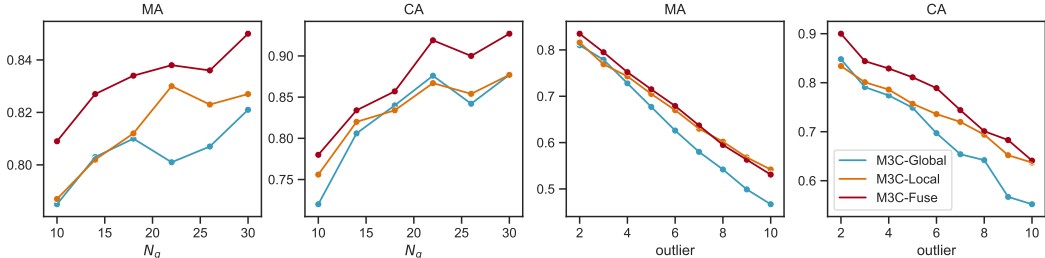

Figure 4: Comparison by four metrics of the three proposed ranking schemes (local, global, and fuse in Section 4.3) on the Willow ObjectClass dataset, by varying the cluster size $N_g$ and number of outliers.

Table 9: Comparison of Spectral Clustering and Multi-Cut on learning-free solvers under the setting of $N_C = 3$, $N_g = 8$, and $n_o = 2$ outliers. Algorithms with '-MC' use multicut in clustering.

| Method | | MA ↑ | CA ↑ | CP ↑ | RI ↑ |
|---|---|---|---|---|---|
| With Spectral Clustering | MGM-Floyd | 0.709 | 0.567 | 0.673 | 0.699 |
| | M3C | 0.687 | 0.653 | 0.750 | 0.758 |
| With Multi-Cut | MGM-Floyd-MC | 0.709 | 0.603 | 0.716 | 0.724 |
| | M3C-MC | 0.687 | 0.634 | 0.734 | 0.745 |

Our findings from Fig. 4 indicate that M3C-Fuse outperforms all other methods, leading us to select M3C-Fuse as the solver for our unsupervised model, UM3C. These results also confirm that both global and local ranking schemes serve as effective approximations. Furthermore, this demonstrates the robustness and generalization ability of our ranking methods. Even in the presence of 10 outliers, our method achieves a matching accuracy exceeding 0.5 and a clustering accuracy surpassing 0.65. Additionally, the performance of our methods improves in both matching and clustering accuracy as the number of graphs increases. This observation also explains why M3C does not outperform other learning-free solvers in Table 2 (in simpler settings) but demonstrates significant superiority in Table 8 (in more complex settings).

### H.3    COMPARISON OF DIFFERENT CLUSTERING ALGORITHMS

In previous experiments, for all solvers (under the settings of 2GM, MGM, and MGMC), we adopt the same procedure for clustering. The first step involves computing the affinity score $J_{ij}$ for each pair of graphs. To sparsify the affinity scores, we select only the 10 nearest neighbors for each graph and mask other pairwise affinities, following the approach in Wang et al. (2020b) to obtain a more effective input matrix. Subsequently, we employ a clustering algorithm on the sparse affinity $\{J_{ij}^k\}$.

We now conduct a comparison between the clustering algorithms Spectral Clustering (Ng et al., 2002) and Multi-Cut (Swoboda & Andres, 2017) applied to two well-established traditional algorithms: MGM-Floyd and M3C-Fuse. We hope this comparison justifies our choice of clustering algorithm.

Table 9 presents the performance of all four combinations. As a result, there is no substantial alternation in clustering performance. As both (Wang et al., 2020b;a) utilized spectral clustering, to ensure a fair comparison, we adhere to their protocol and employ spectral clustering in our primary experiments.

Furthermore, we posit that the key to achieving effective clustering lies in obtaining high-quality matching and forming reliable affinity scores for clustering. Multi-Cut, as well as Spectral Clustering, represents just one approach to produce robust clustering. The clustering visualization of different methods is shown in Fig. 5.

### H.4    CONVERGENCE STUDY OF M3C

We experiment to show the changes in the supergraph structure of M3C-hard, M3C, and DPMC per iteration in Table 10. In the case of the two M3C variants, the structure refers to the cluster indicator (also the corresponding supergraph), whereas for DPMC, it pertains to the designed tree structure.

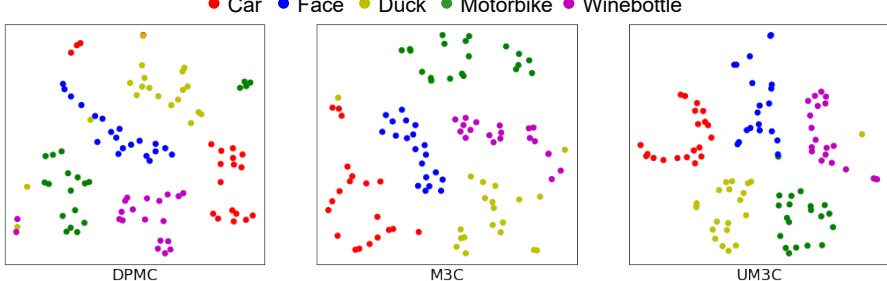

● Car  ● Face  ● Duck  ● Motorbike  ● Winebottle

DPMC                    M3C                    UM3C

Figure 5: Cluster visualization by projecting into 2-D space. We show the spectral embedding of different methods: DPMC, M3C, and UM3C under $5 \times 20, 2$ outliers setting. The embedding is obtained based on pairwise affinity score and the dimension of the embedding space is 16. We apply t-SNE to reduce the dimension to 2 to draw the visualization figures.

Table 10: Changes in supergraph structure (measured by the number of changed edges per iteration $\sum \left| \mathbf{A}^{(t+1)} - \mathbf{A}^t \right|$) over iterations under the setting of $N_c = 3$, $N_g = 8$ with 2 outliers as disturbance. For M3C, the structure is the cluster indicator $\hat{\mathbf{C}}$, and for DPMC, the structure is the maximum spanning tree.

| Iteration # | 2 | 3 | 4 | 5 | 6 | 7 | 8 | 9 | 10 |
|---|---|---|---|---|---|---|---|---|---|
| M3C-hard | 10.48 | 0.56 | 0 | 0 | 0 | 0 | 0 | 0 | 0 |
| M3C | 20.44 | 2.04 | 1.56 | 0.24 | 0.04 | 0.08 | 0 | 0 | 0 |
| DPMC | 10.16 | 6.16 | 3.28 | 1.20 | 0.48 | 0.32 | 0.24 | 0.24 | 0.24 |

The number is the edges changed per iteration, which is calculated by $\sum \left| \mathbf{A}^{(t+1)} - \mathbf{A}^{(t)} \right|$ where $\mathbf{A}$ is the adjacency matrix of the respective supergraph. It is evident that DPMC oscillates without convergence, while M3C-hard converges rapidly to a local optimum, confirming the fact that the cluster size in M3C-hard remains unchanged in most cases in the experiments and our proposed Proposition. 4.1 holds. M3C exhibits a more balanced convergence rate, leading to its well-balanced performance.

Additionally, we validate the convergence of the three M3C variants using different ranking schemes: M3C-Global, M3C-Local, and M3C-Fuse. The experiment is conducted on the Willow ObjectClass dataset, with the settings of $N_c = 5$, $N_g = 20$, and the presence of two outliers. We iterate each algorithm for 6 cycles and report the mean and standard deviation of the curves based on 50 repetitions. The hyperparameters $r$ for M3C-Global, M3C-Local, and M3C-Fuse are set to 0.05, 0.04, and 0.06, respectively. The results are depicted in Fig. 6. They validate that our algorithm achieves rapid convergence within a few iterations. In the case of each algorithm, it attains a near-optimal target score by the second iteration. This supports the earlier assessment of supergraph structure convergence: the second iteration witnesses a significant number of edge changes, which diminishes in the third iteration but still allows room for further enhancement. It is important to note that variations in target scores are a consequence of selecting different values of $r$ for each algorithm.

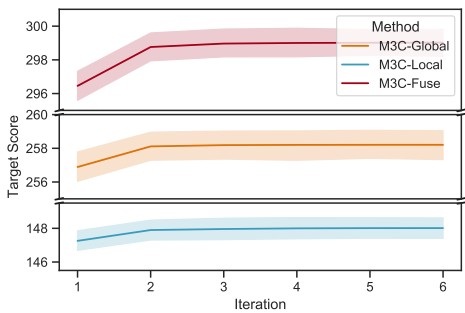

Figure 6: Convergence curve of M3C. The rates are shown under three different schemes of ranking: M3C-Global, M3C-Local, and M3C-Fuse.

## H.5 HYPERPARAMETER STUDY OF M3C

The major hyperparameter for M3C is $r$, which controls the number of graph pairs considered as belonging to the same cluster and determines the number of edges in the supergraph. In this section,

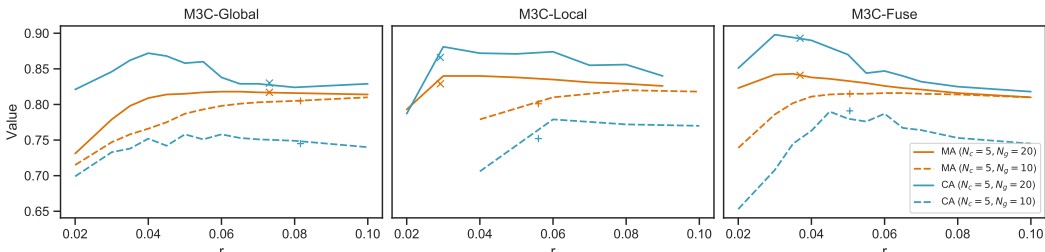

Figure 7: Sensitivity study of the hyperparameter $r$ for the devised three ranking schemes. Experiments are conducted under the setting $N_g = 5$, $N_c = 20$ and $N_g = 5$, $N_c = 10$, both with 2 outliers, on Willow ObjectClass. The marker denotes the performance of our chosen $r$, which is to add edges until the supergraph is connected.

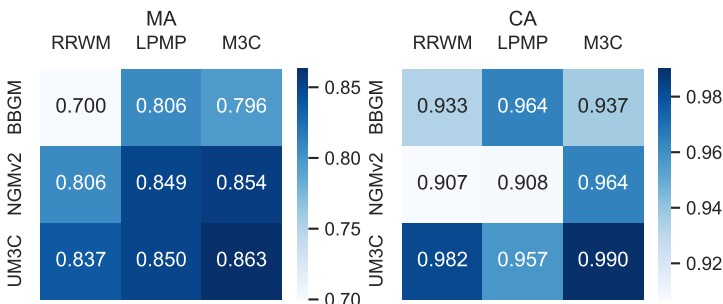

Figure 8: Generalization test of learned affinity under $N_c = 3$ $N_g = 8$, $n_o = 2$ outliers on WillowObject. The x-axis is the solver used, and the y-axis is the learning model of the learned affinity. All models are learned under the same setting as the tests.

we first investigate the sensitivity of the hyperparameter $r$ for M3C-Global, M3C-Local, and M3C-Fuse, and subsequently, we present our tuning algorithm.

Figure 7 illustrates the matching and clustering performance varying the hyperparameter $r$, considering two settings: $N_c = 5$, $N_g = 20$, and $N_c = 5$, $N_g = 10$, each with $n_o = 2$ outliers. It is evident that the matching performance of M3C-Global remains stable when $r > 0.04$ for $N_c = 5$, $N_g = 20$, and $r > 0.06$ for $N_c = 5$, $N_g = 10$. However, its clustering performance deteriorates when $r \geq 0.6$ in both settings. This observation implies that the threshold should be within a reasonable range, as merely adding more edges does not necessarily improve performance. Conversely, having too few edges restricts the algorithm's optimization space. The findings from M3C-Local and M3C-Fuse further support this observation. As depicted in Fig. 7, they achieve optimal results at $r = 0.15$ and $r = 0.03$ for the $5 \times 20$ setting, and $r = 0.3$ and $r = 0.045$ for the $5 \times 10$ setting.

Additionally, it is worth noting that the optimal $r$ varies for different inputs and settings, and determining the best $r$ for each input can be a time-consuming process. Consequently, we employ an alternative approach to address this challenge. Rather than fixing a specific value for $r$, we dynamically add edges based on their rank until the supergraph becomes connected. The symbols '$\times$' and '$+$' in Fig. 7 represent the mean value of $r$ and the corresponding mean performance achieved by this scheme in two settings, respectively. These empirical results demonstrate that this approach provides a reliable approximation of the optimal $r$, enabling the algorithm to attain near-optimal performance without extensive computation. This is the method employed in both our conventional solver, M3C, and the unsupervised learning method, UM3C.

## H.6 GENERALIZATION TEST OF LEARNED AFFINITY $\mathbf{K}^{learn}$

We conducted experiments to evaluate the generalization capability of our learned affinity, demonstrating how our edge-wise affinity loss enhances the robustness of affinity across different solvers. The experiments were conducted on Willow ObjectClass under a $3 \times 8$ setting with 2 outliers, employing the solvers RRWM (Cho et al., 2010) (used in M3C), LPMP (Swoboda et al., 2017) (utilized in BBGM), and M3C, as well as the affinities learned by BBGM, NGMv2, and UM3C. In the case of UM3C, only $\mathbf{K}^{learn}$ was utilized for testing.

As illustrated in Fig. 8, UM3C exhibited superior generalization capabilities in terms of both matching and clustering accuracy. BBGM's learning pipeline limited its applicability to the LPMP solver, offering less utility for other solvers. Furthermore, the affinity generated by UM3C significantly improved the performance of LPMP compared to that generated by BBGM. These results affirm that our edge-wise affinity learning enhances the robustness of the learned affinity, making it adaptable to various solvers.

## H.7 SENSITIVITY TEST OF PSEUDO-LABELS

Table 11: Sensitivity of pseudo-labels for UM3C. Experiments are conducted on Willow Object where we select Car, Duck, and Motorbike as the cluster classes. No outliers are included and we apply RRWM to generate pseudo-labels during the training. "80%RRWM+UM3C" denotes that the pseudo-labels is constructed by 80% RRWM results and 20% random permutation results.

| Model | $N_c$ | $N_g$ | MA↑ | CA↑ | CP↑ | RI↑ | time(s)↓ |
|---|---|---|---|---|---|---|---|
| 100%RRWM + UM3C | 3 | 8 | 0.955 | 0.983 | 0.988 | 0.988 | 3.2 |
| 80%RRWM + UM3C | 3 | 8 | 0.786 | 0.932 | 0.953 | 0.947 | 3.3 |
| 60%RRWM + UM3C | 3 | 8 | 0.546 | 0.86 | 0.902 | 0.886 | 3.1 |
| 40%RRWM + UM3C | 3 | 8 | 0.261 | 0.678 | 0.74 | 0.746 | 3.3 |
| 20%RRWM + UM3C | 3 | 8 | 0.156 | 0.512 | 0.585 | 0.625 | 3.1 |

We conducted an experiment to assess the sensitivity of pseudo-labels. Table 11 reveals that matching accuracy is significantly influenced by the quality of pseudo-labels, demonstrating an almost linear relationship. It is noteworthy that clustering metrics appear more resilient to the degradation of pseudo-labels. Even with a 60% RRWM, the clustering results only decrease by 12.3% on CA, 8.6% on CP, and 10.2% on RI. These findings illustrate that UM3C effectively clusters objects even in scenarios where matching pseudo-labels are lacking. However, Table 11 also indicates that if the pseudo-labels are excessively poor (20% RRWM and 40% RRWM), the cluster results may deteriorate rapidly.

# I   VISUALIZATIONS

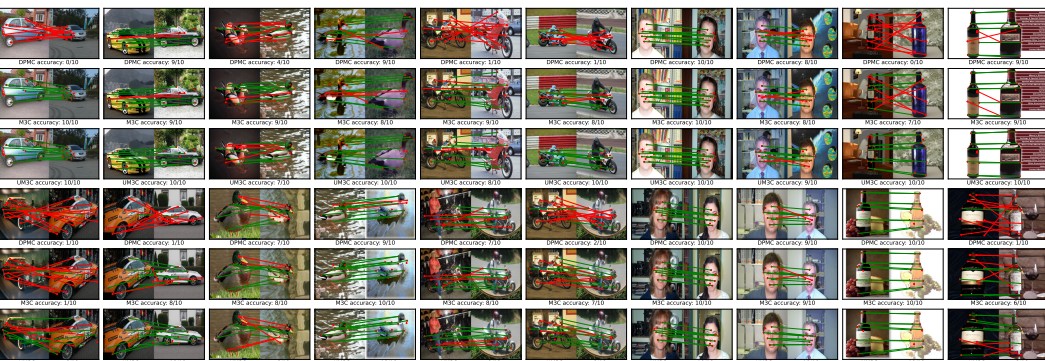

Figure 9: Comparison of different methods: DPMC(top), M3C(Middle), and UM3C(bottom). It is run on the setting with $N_c = 5$ and $N_g = 20$ and 2 outliers. Accuracy is reported for each pairwise matching. All the pairs are randomly picked. Better viewing with color and zooming in.

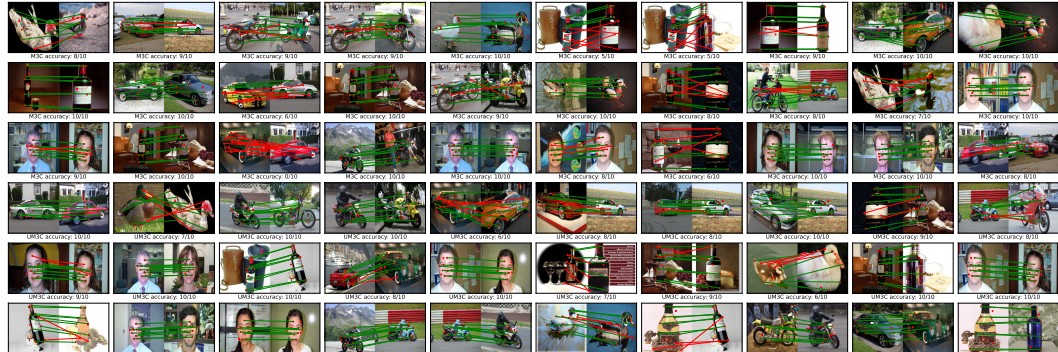

Figure 10: Visualization of our methods: M3C(top) and UM3C(bottom). It is run on the setting with $N_c = 5$ and $N_g = 20$ graphs and 2 outliers. Accuracy is reported for each pairwise matching. All the pairs are randomly picked. Better viewing with color and zooming in.

