# OpenReview forum: "M3C: A Framework towards Convergent, Flexible, and Unsupervised Learning of Mixture Graph Matching and Clustering"
_ICLR.cc/2024/Conference — ICLR 2024 poster_

### Official Review · Reviewer_ra71 · 2023-10-30

**Soundness:** 3 good
**Presentation:** 4 excellent
**Contribution:** 3 good
**Rating:** 8
**Confidence:** 5

**Summary:**

The paper presents a full framework dedicated to graph matching using both unsupervised and supervised methods. The framework mixes graph matching and clustering leading to a method able to deal with heterogeneous set of graphs in a fully unsupervised way. The experiments show promising results on classical datasets.

**Strengths:**

The framework seems very general and address a very classical problem in computer vision. Compared to previous methods it can use both node and edges attributes for matching. The combination of supervised and unsupervised methods helps to improve classical methods with only handcraft features. Most of the paper is clear and easy to read.

**Weaknesses:**

The proposed work has several weakness,

- the pairwise graph matching relies on the Lawler's QAP formulation which is known to not scale well with the size of the graphs. In the experiement all the graphs are shorts (less than 20 nodes for most), this is not the case in general.
- the framework need an initialization with a classical pairwise method. We may expect some sentivity with the chosen method. I did not see any discussion on this part. Furthermore there exists better methods than RRWM like
    - [KerGM](https://proceedings.neurips.cc/paper_files/paper/2019/hash/cd63a3eec3319fd9c84c942a08316e00-Abstract.html) which is able to use edge attributes
    - [GWL](http://proceedings.mlr.press/v97/xu19b.html)
- the comparison with the state of art is missing many others existing methods. The proposed methods are all from the very same team. I would expect a better state-of-art. For example on the deep learning side we have (to cite a few),
    - [SIGMA](https://proceedings.mlr.press/v139/liu21i/liu21i.pdf)
    - [Universe Points Representation Learning for Partial Multi-Graph Matching](https://arxiv.org/abs/2212.00780)
    - [DGMC](https://openreview.net/forum?id=HyeJf1HKvS)
- the full framework is focus on images. It is difficult to assess if it can be extended to general graphs. For example, in the DGMC paper there is an experiment where the attributes are only coordinates.

**Questions:**

I have some questions on the cluster part,
- the MM method asks to solve two problems (namely equations (6) and (7)). Both problems remain hard to solve so I don't see how they can be solved in a proper way. Only one method is proposed but not really described.
- the supergraph is an important tool here. How is it really built? Do we need some heuristic to lessen the problem?

Some other questions on the deep learning part,
- how the features on edges are build? I don't see how the VGG-16 features are used in this case.
- how much the method is sensitive toward the initialization of the pseudo-labels? From the experiments, RRWM seems good enough (in the sense they don't completely failed).

---

> ### Author Response · Authors · 2023-11-16
>
> Thank you for your suggestions regarding the related work. We provide the following response in the hope of resolving your questions.
>
> - **Q1:** the MM method asks to solve two problems (namely equations (6) and (7)). Both problems remain hard to solve so I don't see how they can be solved in a proper way. Only one method is proposed but not really described.
>   - In ensuring a fair comparison with prior work, we employ spectral clustering for Eq 6 and utilize MGM-Floyd as the solver for Eq 7, given its state-of-the-art performance in addressing the multi-graph matching problem. Furthermore, in Appendix H.3, we conduct an experiment demonstrating minimal variance in results when transitioning from spectral clustering to the multi-cut algorithm. This observation underscores our argument that the crux of MGMC lies in the joint solving of the two problems, affirming its pivotal role in the overarching task.
>
> - **Q2:** the supergraph is an important tool here. How is it really built? Do we need some heuristic to lessen the problem?
>   - The concept of a supergraph involves representing all input graphs as nodes, with connectivity and edge weight (often the affinity score) indicating the relationship between the graphs. In typical multi-graph matching algorithms, the supergraph manifests as a complete graph or clique, where each edge's weight corresponds to the affinity score between any two graphs. However, our proposed framework deviates from this norm by constructing a complete graph within clusters and maintaining no edges between them (M3C-Hard).
>   - Proposition 4.1 elucidates that this non-optimal structure, derived from clustering, serves as a critical aspect of our M3C algorithm. To enhance the solution, we introduce three ranking schemes for constructing a supergraph, addressing the limitations of the initial complete graph construction. Thus, the dual contributions of our M3C algorithm lie in its general framework and the innovation of an improved supergraph construction method.
>
> - **Q3:** how the features on edges are build? I don't see how the VGG-16 features are used in this case.
>   -  The edge feature is computed as $\mathbf{E}_{i,j}=\mathbf{F}^n_i - \mathbf{F}^n_j$ in practice to align with prior work.
>
> - **Q4:** how much the method is sensitive toward the initialization of the pseudo-labels? From the experiments, RRWM seems good enough (in the sense they don't completely failed).
>   - We performed a sensitivity experiment on pseudo-labels for UM3C. The experiments were carried out on the Willow Object dataset, where we selected Car, Duck, and Motorbike as the cluster classes. Outliers were not included, and RRWM was applied to generate pseudo-labels during training. The term "80% RRWM + UM3C" indicates that the pseudo-labels are constructed using 80% RRWM results and 20% random permutation results. The results are presented below. Importantly, it is observed that clustering metrics exhibit greater resilience than matching accuracy to the degradation of pseudo-labels. For further discussion, please refer to Appendix H7.
> | Model            | $N_c$ | $N_g$ | MA | CA| CP | RI | time(s) |
> |------------------|-------|-------|--------------|--------------|--------------|--------------|---------------------|
> | 100\%RRWM + UM3C | 3     | 8     | 0.955        | 0.983        | 0.988        | 0.988        | 3.2                 |
> | 80\%RRWM + UM3C  | 3     | 8     | 0.786        | 0.932        | 0.953        | 0.947        | 3.3                 |
> | 60\%RRWM + UM3C  | 3     | 8     | 0.546        | 0.86         | 0.902        | 0.886        | 3.1                 |
> | 40\%RRWM + UM3C  | 3     | 8     | 0.261        | 0.678        | 0.74         | 0.746        | 3.3                 |
> | 20\%RRWM + UM3C  | 3     | 8     | 0.156        | 0.512        | 0.585        | 0.625        | 3.1                 |

---

> ### Author Response · Authors · 2023-11-16
>
> - **W1:** the pairwise graph matching relies on the Lawler's QAP formulation which is known to not scale well with the size of the graphs. In the experiement all the graphs are shorts (less than 20 nodes for most), this is not the case in general.
>   - PascalVOC serves as a challenging benchmark even in the two-graph matching setting with 10-20 key points. Notably, state-of-the-art two-graph matching methods achieve only ~83% matching accuracy (full matching) and ~71% F1 score (for partial matching) on this dataset. Though achieving state-of-the-art performance on MGMC, we believe that these datasets could still serve as the benchmark given the space for improvement. When an algorithm eventually solve the problem on this dataset (e.g. over 90 matching accuracy), this field can move into larger graphs with more key points and more complex structure.
>
> - **W2:** the comparison with the state of art is missing many others existing methods. The proposed methods are all from the very same team. I would expect a better state-of-art. For example on the deep learning side we have (to cite a few),
>   - We have added all the mentioned papers into related work.
>   - We also add experiments of two latest works: GCAN[1] and COMMON[2] on PascalVOC with 3*8 setting, where COMMON is the sota of 2GM setting on PascalVOC. Here, only a partial presentation of the results is provided. The complete set of comparison experiments has been incorporated into Section 6 (Experiments), and the corresponding discussion has been revised accordingly. For more details, we kindly request you to refer to the complete paper.
> | Setting | Protocal     | Model      | MA     | CA     | CP     | RI     | time   |
> | ------- | ------------ | ---------- | ------ | ------ | ------ | ------ | ------ |
> | 3*8     | supervised   | NGMv2      | 0.8114 | 0.755  | 0.8083 | 0.8165 | 4.2586 |
> | 3*8     | supervised   | BBGM       | 0.7919 | 0.7973 | 0.8406 | 0.8371 | 2.2618 |
> | 3*8     | supervised   | UM3C+BBGM  | 0.8037 | 0.8945 | 0.9212 | 0.918  | 5.1646 |
> | 3*8     | supervised   | GCAN       | 0.8049 | 0.8089 | 0.8537 | 0.8438 | -      |
> | 3*8     | unsupervised | GCAN+M3C   | 0.75   | 0.824  | 0.8637 | 0.8565 | -      |
> | 3*8     | supervised   | COMMON     | 0.8334 | 0.9318 | 0.9467 | 0.9458 | -      |
> | 3*8     | unsupervised | COMMON+M3C | 0.8435 | 0.9494 | 0.9629 | 0.9595 | -      |
>
> - **W3:** the full framework is focus on images. It is difficult to assess if it can be extended to general graphs. For example, in the DGMC paper there is an experiment where the attributes are only coordinates.
>   - Actually, in the learning-free setting, only coordinates are used as input to calculate the hand-crafted affinity K, without involving the features of image points. Our learning-free method M3C demonstrates its competitiveness compared to other learning-free algorithms in that case.
>
> [1] Zheheng Jiang, Hossein Rahmani, Plamen Angelov, Sue Black, Bryan M. Williams. "Graph-Context Attention Networks for Size-Varied Deep Graph Matching." CVPR 2022
>
> [2] Yijie Lin, Mouxing Yang, Jun Yu, Peng Hu, Changqing Zhang, Xi Peng. "Graph Matching with Bi-level Noisy Correspondence." ICCV 2023

---

> > ### Comment · Reviewer_ra71 · 2023-11-22
> > **Response to the authors**
> >
> > Many thanks for the clarification. From the the answers of the other reviews, I have now a better overview of the pro/cons of the methods. While I still think that comparison against older but interesting methods would clearly strengthen the experiments, I also understand that it is hard to launch all of these in a short time. Since the main weakness are clarified I will rise my rating to 8.

---

### Official Review · Reviewer_sBU3 · 2023-10-30

**Soundness:** 3 good
**Presentation:** 3 good
**Contribution:** 3 good
**Rating:** 8
**Confidence:** 3

**Summary:**

This work explores a practical scenario in graph matching, where the collected graphs are of different types. To tackle this issue, the authors introduced a strategy named MGMC which simultaneously performs graph clustering and graph matching, along with a novel method M3C as an implementation of such strategy. M3C not only handles graph matching with mixed types, but also addresses several drawbacks of previous graph matching methods. Overall, the studied topic is meaningful and the work is solid despite some minor weaknesses, so I recommend acceptance for presenting it at ICLR.

**Strengths:**

* The paper is easy to read, and well-written in general.
* The studied scenario in which the dataset is a mixture of different graph types is important in practical applications.
* The proposed model along with its MM-based optimization algorithm solves several drawbacks of previous graph matching methods.

**Weaknesses:**

* The literature part lacks state-of-the-art works published in the last two years.
* The authors didn't compare with the latest works. The most recent competitor MGM-Floyd was published in 2021.

**Questions:**

* What's the major benefit of MGMC? For matching with mixed graph types, we could employ graph-level classification or clustering (so that labeling is also avoided) methods to preprocess the dataset and apply conventional graph matching methods to individual classes. No ablation experiment is conducted to verify the effectiveness of MGMC anyway.
* Is it possible to integrate the proposed method into a fully end-to-end GM pipeline (such as NGMv2)?

---

> ### Author Response · Authors · 2023-11-16
>
> - Thanks for your time and valuable comments. Below we respond to your concerns and hope we can clarify some questions.
>
> - **Q1:** What's the major benefit of MGMC? For matching with mixed graph types, we could employ graph-level classification or clustering (so that labeling is also avoided) methods to preprocess the dataset and apply conventional graph matching methods to individual classes. No ablation experiment is conducted to verify the effectiveness of MGMC anyway.
>   - The principal advantage of MGMC lies in its capacity to mutually optimize matching and clustering, enhancing the outcomes of both tasks.
>   - Clustering benefits from graph matching as a similarity metric, as highlighted by Wang et al., 2020a, who note, "The key challenge of multi-graph clustering is finding a reasonable measurement for graph-wise similarity, after which the common spectral clustering technique can be applied to discover clusters. We tackle this problem by proposing a matching-based graph-wise similarity measure for clustering. "
>   - Conversely, clustering also contributes to improved matching results. The distinction between multi-graph matching and two-graph matching lies in cycle consistency, necessitating transitivity across multiple graphs. However, this consistency becomes irrelevant between graphs of different clusters. Therefore, clustering provides a strategic division for graphs, allowing focused attention on cycle consistency within clusters while leaving other graphs unaffected.
>   - As noted by the reviewer, the suggestion to use graph-level classification or clustering methods for dataset preprocessing aligns with a one-iteration M3C-hard approach. This involves clustering the graphs initially and subsequently applying the MGM algorithm to each cluster.
>
>
> - **Q2:** Is it possible to integrate the proposed method into a fully end-to-end GM pipeline (such as NGMv2)?
>     - Yes. It is feasible to integrate the method into a fully end-to-end GM pipeline. In Table 2, we have showcased an end-to-end unsupervised learning version of our method, which demonstrates superior performance compared to the unsupervised learning method GANN. Furthermore, by enhancing the feature extractor through our proposed affinity learning and employing our M3C solver for refining two-graph matching, a supervised end-to-end learning pipeline for M3C can be established based on NGMv2.
>
>
> - **W1:** The literature part lacks state-of-the-art works published in the last two years.
>   - We have added GCAN[1], COMMON[2], Universe Points Representation Learning for Partial Multi-Graph Matching[3] to our related work.
>
> - **W2:** The authors didn't compare with the latest works. The most recent competitor MGM-Floyd was published in 2021.
>   - We add two latest works: **GCAN**[1] and **COMMON**[2] on PascalVOC with 3*8 setting, where **COMMON** is the sota of 2GM setting on PascalVOC. Here, only a partial presentation of the results is provided. The complete set of comparison experiments has been incorporated into Section 6 (Experiments), and the corresponding discussion has been revised accordingly. For more details, we kindly request you to refer to the complete paper.
> | Setting | Protocal     | Model      | MA     | CA     | CP     | RI     | time   |
> | ------- | ------------ | ---------- | ------ | ------ | ------ | ------ | ------ |
> | 3*8     | supervised   | NGMv2      | 0.8114 | 0.755  | 0.8083 | 0.8165 | 4.2586 |
> | 3*8     | supervised   | BBGM       | 0.7919 | 0.7973 | 0.8406 | 0.8371 | 2.2618 |
> | 3*8     | supervised   | UM3C+BBGM  | 0.8037 | 0.8945 | 0.9212 | 0.918  | 5.1646 |
> | 3*8     | supervised   | GCAN       | 0.8049 | 0.8089 | 0.8537 | 0.8438 | -      |
> | 3*8     | unsupervised | GCAN+M3C   | 0.75   | 0.824  | 0.8637 | 0.8565 | -      |
> | 3*8     | supervised   | COMMON     | 0.8334 | 0.9318 | 0.9467 | 0.9458 | -      |
> | 3*8     | unsupervised | COMMON+M3C | 0.8435 | 0.9494 | 0.9629 | 0.9595 | -      |
>
> [1] Zheheng Jiang, Hossein Rahmani, Plamen Angelov, Sue Black, Bryan M. Williams. "Graph-Context Attention Networks for Size-Varied Deep Graph Matching." CVPR 2022
>
> [2] Yijie Lin, Mouxing Yang, Jun Yu, Peng Hu, Changqing Zhang, Xi Peng. "Graph Matching with Bi-level Noisy Correspondence." ICCV 2023
>
> [3] Nurlanov Z, Schmidt F R, Bernard F. Universe points representation learning for partial multi-graph matching[C]//Proceedings of the AAAI Conference on Artificial Intelligence. 2023, 37(2): 1984-1992.

---

### Official Review · Reviewer_euAu · 2023-10-30

**Soundness:** 2 fair
**Presentation:** 1 poor
**Contribution:** 2 fair
**Rating:** 6
**Confidence:** 4

**Summary:**

This paper proposes a novel algorithm to jointly optimize graph clustering and matching. The optimization problem is formulated as a minorize-maximization problem with guaranteed convergence. An unsupervised variant is further introduced to incorporate edge-wise affinity and pseudo label selection. Experiments demonstrate the effectiveness of the proposed method.

**Strengths:**

-	It is a novel idea to jointly address graph clustering and matching problems simultaneously, which are mutually beneficial.
-	Fast convergence is achieved by the minorize-maximization framework.

**Weaknesses:**

-	The presentation of the paper needs further improvements. Several confusing symbols/notations are abused or used w/o declaration.
-	The theoretical analysis may need further justification.
-	Though the UM3C method is called to be unsupervised, it still requires the hand-crafted affinity matrix, which is another kind of supervision.

**Questions:**

-	Section 4.1: what’s the definition for $g(X|X^t)$? Is it the graph matching objective function given the clustering result h(X^t)?
-	Convergence statement above Eq.(3): Eq.(3) only guarantees your objective function is non-decreasing, but not necessarily guarantee convergence? I think another important reason for the convergence of f is that the solution space (X,C) is finite. In some scenarios, even the objective function remains unchanged, multiple optimal solutions may exists (i.e., $f(X_i,C_i)=f(X_j,C_j)$), and the solution may switch between $(X_i,C_i$ and $X_j,C_j$) instead of converging.
-	Proposition 4.1: $N_{g_i}$ are used w/o definition. How can you guarantee the if condition that the sizes of clusters are the same in two consecutive iterations? I think this is a quite strict condition, and hence I don’t think this proposition provide insightful understanding to the convergence. Besides, you claimed a ‘quick convergence’ in Appendix C.3, it’s necessary to provide a convergence rate.
-	Section 4.2: you claimed disregarding the transitive relations as an advantage, can you explain why? As mentioned later you adopt a greedy search to get the top-$ rN^2$ values, this may breaks the transitive constraint, making it possible to have c_{ik}=1,c_{kj}=1$ but $c_{ij}\neq 1$, which does not make sense to me.
-	Eq. (6): how are $r$ selected? I think this is highly heuristic and may dramatically influence the model performance.
-	Eq. (8): you claimed your method as unsupervised, but you need the hand-crafted $K^{raw}$ as input, which actually is another kind of supervision.
-	Writing
  - Definition 2: $k_1,k_2,…$ and $N$ used w/o definition; How can you multiply the two matrices $X_{ik_1}$ with $X_{k_1k_2}$ as they are not necessarily (and mostly) with the same shape? The vertex set $\mathcal{V}$ is defined as a set of graphs.
  - Eq.(18): $c_{ij}^t\to c_{ij}^{(t)}$.
  - For equations not quoted in the main text, you should not number it and use ‘equation*’ environment instead.

---

> ### Author Response · Authors · 2023-11-16
>
> Thanks for your time and valuable comments. Below we respond to your concerns and hope we can clarify some questions.
> - **Q1:** What’s the definition for $g(X|X^t)$?Is it the graph matching objective function given the clustering result h(X^t)?
>   - Yes, $g(X|X^t) = F(X, h(X^t))$ is a surrogate function crafted within the standard MM framework. By optimizing this surrogate function, we effectively ascertain the optimal matching result $X$ while keeping the cluster division $h(X^t)$ fixed. The construction of the surrogate function $g(X|X^t)$ is aimed at avoiding mutual optimization between matching and clustering, thereby simplifying the overall optimization task.
>
> - **Q2:** Convergence statement above Eq.(3): Eq.(3) only guarantees your objective function is non-decreasing, but not necessarily guarantee convergence? I think another important reason for the convergence of f is that the solution space (X,C) is finite. In some scenarios, even the objective function remains unchanged, multiple optimal solutions may exists (i.e., $f(X_i,C_i)=f(X_j,C_j)$), and the solution may switch between $(X_i,C_i$ and $X_j,C_j$) instead of converging.
>   - Allow us to elaborate on the convergence theorem underpinning this statement. MGMC problem inherently resembles an assignment problem, and its objective function exhibits a natural upper bound. For example, the objective function is smaller than the sum of all the affinity matrices for all assignments:$f(X^t) < \sum_{ij} \sum_{abcd} K_{ij}(a,b,c,d) \forall X^t$ .
>   - According to the **Monotone Convergence Theorem**, if a monotone sequence of real numbers, such as $f(X^t)$, is bounded (possessing an upper bound for an increasing sequence or a lower bound for a decreasing sequence), then the sequence converges.
>   - Evidently, $f(X^t)$ is non-decreasing and bounded, satisfying the conditions for convergence. This convergence holds true even if we relax the solution space $(X, C)$ into a continuous space since the function remains bounded.
>
> - **Q3:** Proposition 4.1: $N_{g_i}$ are used w/o definition. How can you guarantee the if condition that the sizes of clusters are the same in two consecutive iterations? I think this is a quite strict condition, and hence I don’t think this proposition provide insightful understanding to the convergence. Besides, you claimed a ‘quick convergence’ in Appendix C.3, it’s necessary to provide a convergence rate.
>   - $N_{g_i}$ refers to the graph number of the i-th cluster, as defined in Appendix A. Due to the page limitation, we listed all definitions in Appendix A Table 3.
>   - The proposition 4.1 is presented as an extreme case demonstrating the possibility of quick convergence within a single optimization step. However, it is often the case in real world experiment. **Throughout the convergence experiments in Appendix H.4 , the cluster size in M3C-Hard remained unchanged in most cases**, aligning with the condition that sizes of clusters are the same.
>   - The quick convergence attributed to hard clustering arises from the optimization of the surrogate function $g(X|X^t)$, where the optimization process encourages graphs within the same cluster to move closer to each other.
>   - To elaborate, in the optimization of $g(X|X^t)$, the formulation involves maximizing the similarity within clusters, as indicated by the term
> $$\frac{1}{\sum_{ij} c_{ij}} \sum_{ij} c_{ij} \text{vec}(X_{ij})^{T} K_{ij} \text{vec}(X_{ij}),$$
> with $C=h(X^{t}$) fixed. The binary nature of $c_{ij}$ (0 or 1) implies that only pairs within the same cluster undergo optimization. Consequently, graphs within the same cluster are encouraged to converge. This characteristic of hard clustering expedites convergence but limits the ability to correct clustering results during the optimization process.
>   - Furthermore, in Appendix H.4 Table 10, we present convergence experiments showcasing changes in the supergraph structure of M3C-hard, M3C, and DPMC per iteration. When the supergraph structure stabilizes, the optimization is fixed and the results converge. As illustrated in Table 10 (also shown below), M3C-Hard achieves swift convergence in merely three steps.
> | Iteration | 2 | 3 | 4 | 5 | 6 | 7 | 8 | 9 | 10 |
> | --- | --- | --- | --- | --- | --- | --- | --- | --- | --- |
> | M3C-Hard | 10.48 | 0.56 | 0 | 0 | 0 | 0 | 0 | 0 | 0 |
> | M3C | 20.44 | 2.04 | 1.56 | 0.24 | 0.04 | 0.08 | 0 | 0 | 0 |
> | DPMC | 10.16 | 6.16 | 3.28 | 1.20 | 0.48 | 0.32 | 0.24 | 0.24 | 0.24 |

---

> ### Author Response · Authors · 2023-11-16
>
> - **Q4:** Section 4.2: you claimed disregarding the transitive relations as an advantage, can you explain why? As mentioned later you adopt a greedy search to get the top-$ rN^2$ values, this may breaks the transitive constraint, making it possible to have $c_{ik}=1,c_{kj}=1$ but $c_{ij}\neq 1$, which does not make sense to me.
>   - The hard clustering division supergraph and fully connected supergraph represent two extrem cases for solving MGMC problem. The former eliminates the influence of graph pairs from other clusters, even if the cluster division is not precisely correct. On the other hand, the fully connected supergraph encompasses an excessively large solution space, laden with invalid information. Our relaxation strategy functions as a compromise between these two extremes. It expands the optimization space compared to hard clustering division, considering graphs that may have been wrongly assigned during the matching step. However, compared to the fully connected supergraph, it introduces restrictions to mitigate the influence of graph pairs from different clusters.
>   - Now going back to your question about breaking the transitive constraint, the strict enforcement of the transitive constraint implies that clusters must be independent in the supergraph, and the supergraph inside each cluster must form a clique. Given that the clustering result could be erroneous, this constraint could potentially hinder the utilization of useful information from other clusters, even if it was incorrectly assigned. Relaxing this constraint becomes a necessary step to achieve better performance within our framework. The relaxation enables the model to consider possibly valuable information from other clusters, improving the adaptability of the algorithm while pertaining to the nature of clustering.
>
> - **Q5:** Eq. (6): how are $r$ selected? I think this is highly heuristic and may dramatically influence the model performance.
>   - We discussed the influence of hyperparameters in Appendix H.5. The selection of $r$ in Eq. (6) and its potential impact on model performance is indeed acknowledged. It's crucial to recognize that the optimal $r$ is not universally applicable across different inputs and settings. Determining the optimal $r$ for each input can be a time-consuming process.
>   - To address this challenge, we adopt an alternative strategy. Instead of fixing a specific value for $r$, we dynamically add edges based on their rank until the supergraph is connected. This adaptive approach aims to circumvent the need for manual tuning and accommodates variations in input characteristics. A comparison between this dynamic edge addition approach and fixing a specific $r$ is presented in Appendix H.5 Figure 7.
>
> - **Q6:** Eq. (8): you claimed your method as unsupervised, but you need the hand-crafted $K^{raw}$ as input, which actually is another kind of supervision.
>   - It's important to clarify that the hand-crafted $K^{raw}$ is **not** provided as external supervision. Instead, it is calculated solely based on the input data. As detailed in Appendix F.2, $K^{raw}$ is computed from key point coordinates, incorporating factors such as length and angle. Key point coordinates are fundamental inputs for any graph matching model/algorithm, akin to pixel intensity for image segmentation. Therefore, the inclusion of key point coordinates in the computation of $K^{raw}$ should not be misconstrued as an additional form of supervision. It remains intrinsic to the unsupervised nature of our method.

---

> ### Comment · Reviewer_euAu · 2023-11-21
>
> Thanks for your clarifications, which address most of my concerns. I've also read through discussions from other reviewers, and I believe the clarifications and additional experiments carried out in the rebuttal should be included in the future versions. I would like to increase my rating to 6.

---

> > ### Author Response · Authors · 2023-11-21
> >
> > We want to express our gratitude for your insightful feedback on our paper and willingness to increase the rating to 6. We have carefully considered your suggestions and made the necessary revisions to enhance the quality of our work. The revised version of the paper has been uploaded.

---

### Official Review · Reviewer_uZV9 · 2023-10-31

**Soundness:** 3 good
**Presentation:** 2 fair
**Contribution:** 3 good
**Rating:** 6
**Confidence:** 3

**Summary:**

The author(s) introduce a manner to jointly solve the problem of graph matching and graph clustering. For, they propose an objective function and develop an algorithm in the framework of Minorize-Maximization. The resulting method is called M3C, after adding a relaxation of the hard cluster assignment. The last element of the work is to embed the M3C method into a learning framework for the affinity matrices, so that the method becomes unsupervised.  Numerical experiments study the usefulness of the proposed method.

The contributions are :
i) the proposition of the joint matching and clustering for graphs, and that is interesting
ii) the algorithm to solve that, involving MM, then relaxation of the hard assignment matrix, and then the plugging into deep learning methods so as to obtain an unsupervised version.

All that is moderately original, still it has the advantage that it appears to work.

**Strengths:**

- A sound problem, to try to study jointly matching and clustering for graphs

- The theoretical parts, often postponed in the supplementary sections, are well written. Yet they are coming straight from classical results.

- The numerical experiments are well conducted.

- Adequate numerical performance on the two used datasets for comparison to other methods

- A large choice of numerical  experiments, both in the main text and in the appendices

**Weaknesses:**

- The problem is sound, yet it does not appear to be really important. In addition, I am not certain of the added value of solving the two problems at the same time ; the authors should spend more energy to convince the readers of that. For instance, it seems that frameworks coming from optimal transport for graphs would solve both matching and clustering for graphs. Or isn't it possible ? Why is it better to design a method focusing on both in 1 step ?

- In the absence of an insightful discussion (or proof...) about the necessity of considering graph matching and graph clustering in a joint approach, the article is over-stating somehow its contribution ; even if UM3C works, it's not certain it is needed.

- the criteria proposed in equation (2) appear to be ad hoc

- the Minorize-Maximization (MM) framework is not new, and here used in a classical way (hence most of it would move to the appendices)

- the writing could be improved.
The presentation is nearly adequate, yet it comes with some repetitions.
The derivation of the overall objective for joint matching and clustering, eq (2), is not well presented and one does not fully know where it comes from, if there could be other choices, and globally what is ad hoc in this formulation and what is mandatory.

- The authors rely heavily on the figures (1 and 2) for the readers to understand the full picture, but for me they are not that clear because there are too many elements displayed at the same place.

**Questions:**

- The topic of joint matching and clustering for graphs is sound, although the authors should strive to find situations that are more elaborated than their examples on images. This assumes that images are best coded as graphs, yet nothing is said in support of that; also; the graphs representing images are quite simple.
The authors should think about situations where the affinity graphs are less simple, and the graphs more complex.

- page 1 : K (the affinity matrix) is not defined

- p 6 in 5.1: Where is \Lambda defined ?

- Is there a comparison to simpler graph matching methods, for instance using optimal transport distance, that are known to also be usable for clustering ?

- In 4.2, and then in 4.3, a softened version of the hard assignment matrix, C, is introduced. Is it only anrelaxation for easier convergence, or does the relaxed matrices capture something about possible confusion between the clusters ? It would be useful if it is the case,  as we often have clusters which are not as clear  cut as hard assignment is assuming.

---

> ### Author Response · Authors · 2023-11-16
>
> Thank you for your recognition of the sound problem formulation, well-written theoretical sections, and well-conducted experiments. Below we respond to your concerns and hope we can clarify some questions.
>
> - **Q1:** The topic of joint matching and clustering for graphs is sound, although the authors should strive to find situations that are more elaborated than their examples on images. This assumes that images are best coded as graphs, yet nothing is said in support of that; also; the graphs representing images are quite simple. The authors should think about situations where the affinity graphs are less simple, and the graphs more complex.
>   - We appreciate your recognition of the validity of our joint matching and clustering problem.  We have introduced the concept in the context of traffic tracking, where surveillance camera footage captures diverse entities such as people, bicycles, and cars concurrently. Each of these entities can be delineated by several feature points of interest, forming a complex graph structure. In this scenario, the MGMC paradigm proves invaluable for efficiently tracking different modes of data. Noteworthy applications of graph matching methods in tracking tasks, exemplified by works such as He et al.'s "Learnable Graph Matching: Incorporating Graph Partitioning with Deep Feature Learning for Multiple Object Tracking" (CVPR 2021) and Wu et al.'s "Multiview Vehicle Tracking by Graph Matching Model" (CVPR AI City Challenge 2019), further underscore the versatility of our approach.
>   - However, it is crucial to emphasize that in this paper, we position MGMC more as an **upstream problem**, focusing on the general setting and adherence to standard evaluation benchmarks. We acknowledge that we are still at a nascent stage of the MGMC problem, with only two existing works, DPMC and GANN, addressing it (another relavant work Krahn et al. "Convex Joint Graph Matching and Clustering via Semidefinite Relaxations" adopts a different setting). **Our evaluation aligns with the standard presented in these two papers, advancing them by utilizing a more complex dataset, PascalVOC.** PascalVOC serves as a challenging benchmark even in the two-graph matching setting with 10-20 key points. Notably, state-of-the-art two-graph matching methods achieve only ~83% matching accuracy (full matching) and ~71% F1 score (for partial matching) on this dataset. Though achieving state-of-the-art performance on MGMC, we believe that these datasets could still serve as the benchmark given the space for improvement. When an algorithm eventually solve the problem on this dataset (e.g. over 90 matching accuracy), this field can move into larger graphs with more key points and more complex structure.
>
> - **Q2:** page 1 : K (the affinity matrix) is not defined
>   - $K_{ij} \in R^{n_i\times n_j, n_i\times n_j}$ is the affinity matrix whose diagonal elements and off-diagonal ones encode the node-to-node and edge-to-edge affinity between two graphs $G_i$ and $G_j$, respectively. We give all the definition in Appendix A due to the limitation of pages.
>
> - **Q3:** p 6 in 5.1: Where is \Lambda defined ?
>   - $\Lambda \in R^{d\times d}$ where d is the dimension of the learned feature. $\Lambda$ is a learnable parameter to weight features that is a common practice in matching tasks since the paper "Deep Learning of Graph Matching", A Zanfir et. al, CVPR 2018.
>
> - **Q4:** Is there a comparison to simpler graph matching methods, for instance using optimal transport distance, that are known to also be usable for clustering ?
>   - A prior study by Wang et al. ("Learning Combinatorial Embedding Networks for Deep Graph Matching," ICCV 2019) introduced the PCA method. Operating within a simplified context, PCA ignores edge constraints during matching and employs **Sinkhorn as a differentiable solver, aligning with the concept of optimal transport**. We present PCA's performance in tackling the MGMC problem on the PascalVOC dataset. Our analysis shows that PCA lags behind methods rooted in more complex settings where the edge information is consicdered, such as BBGM and NGMv2. Additionally, PCA demonstrates significantly lower accuracy compared to our UM3C(+BBGM). We believe this assessment adequately addresses the performance of optimal transport-related methods on this particular topic.
>   | Setting | Protocal   | Model     | MA     | CA     | CP     | RI     | time   |
>   | ------- | ---------- | --------- | ------ | ------ | ------ | ------ | ------ |
>   | 3*8     | supervised | PCA       | 0.7228 | 0.7152 | 0.769  | 0.7865 | -      |
>   | 3*8     | supervised | NGMv2     | 0.8114 | 0.755  | 0.8083 | 0.8165 | 4.2586 |
>   | 3*8     | supervised | BBGM      | 0.7919 | 0.7973 | 0.8406 | 0.8371 | 2.2618 |
>   | 3*8     | supervised | UM3C+BBGM | 0.8037 | 0.8945 | 0.9212 | 0.918  | 5.1646 |

---

> ### Author Response · Authors · 2023-11-16
>
> - **Q5:** In 4.2, and then in 4.3, a softened version of the hard assignment matrix, C, is introduced. Is it only anrelaxation for easier convergence, or does the relaxed matrices capture something about possible confusion between the clusters ? It would be useful if it is the case, as we often have clusters which are not as clear cut as hard assignment is assuming.
>   - The incorporation of the softened version of constraint matrix C serves the purpose of guiding the algorithm towards an **improved solution** (as opposed to merely easier convergence) in comparison to the baseline method M3C-Hard. In fact, M3C-Hard converges faster (but to a worse result) as we presented in Appendix H.4 Table. 10.
>   - The efficacy of achieving a better solution can be attributed to the method employed in constructing matrix C, which utilizes three proposed ranking schemes. This comprehensive approach enables the identification and capture of potential confusion between clusters, thereby enhancing the algorithm's capacity to converge towards better solutions.

---

> ### Comment · Reviewer_uZV9 · 2023-11-22
>
> I have read the answers of the authors, as well as the other reviews. The additional clarification and, more impressively, the complements for the numerical experiments are very good. This answers well to the questions that were raised; I am happy to raise my Rating by one level.

---

### Author Response · Authors · 2023-11-21

We sincerely appreciate the dedicated time and effort invested by the reviewers in providing comprehensive and insightful feedback on our submission.

The reviewers have provided commendable feedback on various aspects of our paper. They acknowledge the novelty and significance of our chosen task, deeming it a "novel idea" (euAu) and emphasizing its "importance" (sBU3). Regarding our method, reviewers highlight the "achievement of fast convergence" (euAu), the "resolution of several drawbacks of previous works" (sBU3), and the recognition of our proposed method as a "general framework" (ra71), with well-conducted experiments (uZV9). Moreover, the writing is well-received, particularly the well-written theoretical sections (uZV9) and the overall clarity and readability (ra71 and sBU3).

Concurrently, the reviewers have furnished us with a substantial amount of valuable feedback. Drawing upon these suggestions, we have diligently implemented several modifications to the paper, summarizing in three main aspects. **All the modifications are highlighted in red**.
1. Introduction to the Task and Mutual Optimization (uZV9 and sBU3) : We improved the exposition of the MGMC task background, clarifying the rationale behind this task and the benefits of mutual optimization in Sec. 1 Introduction. The revised statement articulates, "This task aims to mutually optimize both matching and clustering problems, thereby enhancing the outcomes of both tasks: matching establishes a similarity metric for clustering, while cluster information improves the results of intra-cluster matching."
2. Improvements of Presentation in theoretical analysis (Sec. 4.1-4.2) (euAu): We refined the theoretical aspects, including explanations of convergence and a detailed discussion on the drawbacks of hard clustering. The former eliminates the influence of graph pairs from other clusters, even if the cluster division is not precisely correct, thus correcting clustering results more effectively.
3.  Expansion of Experiments (sBU3 and ra71): We introduced additional experiments and baselines, including the latest GCAN and COMMON baselines, further demonstrating the capabilities of UM3C (in Sec 6 Experiments). We emphasized UM3C's adaptability to integrate seamlessly into any model for end-to-end training. For baselines without publicly available code, appropriate citations were added in Section 2 (Related Works). We also conducted experiments on the sensitivity of pseudo-labels, demonstrating clustering metrics' greater resilience to pseudo-label degradation (Appendix H.7).

In our detailed responses to each reviewer, we have meticulously addressed every specific question and concern raised. We hope our responses effectively address the reviewers' concerns and anticipate additional feedback from the AC and reviewers.

---

### Meta-Review · Area_Chair_nfB7 · 2023-12-04

**Metareview:**

This is a well written paper for graph matching that makes use of the MM framework and relaxed clustering. The reviewers appreciated the thorough responses though some addressed concerns about limited novelty/importance of the problem. The authors may be sure to highlight the importance of the contributions in the final version, though the method is correct and well presented. I would also suggest that they cite the relevant literature on MM, as it is not quite as standard in this venue. At the very least, they may cite some of the references or tutorials by K. Lange; there are several recent works applying MM to clustering in machine learning venues, ie Xu and Lange 2019, and many recent works from various groups applying MM to machine learning tasks more broadly (ie Mairal 2015)

**Justification For Why Not Higher Score:**

Limited impact and importance of problem but solid paper

**Justification For Why Not Lower Score:**

Reviewers all lean positive and well written paper

---

### Decision · Program_Chairs · 2024-01-16

Accept (poster)